# Efficient Testable Learning of Halfspaces
# with Adversarial Label Noise

**Ilias Diakonikolas**
University of Wisconsin, Madison
`ilias@cs.wisc.edu`

**Daniel M. Kane**
University of California, San Diego
`dakane@ucsd.edu`

**Vasilis Kontonis**
University of Texas, Austin
`mailto:vasilis@cs.utexas.edu`

**Sihan Liu**
University of California, San Diego
`il046@ucsd.edu`

**Nikos Zarifis**
University of Wisconsin, Madison
`zarifis@wisc.edu`

## Abstract

We give the first polynomial-time algorithm for the testable learning of halfspaces in the presence of adversarial label noise under the Gaussian distribution. In the recently introduced testable learning model, one is required to produce a tester-learner such that if the data passes the tester, then one can trust the output of the robust learner on the data. Our tester-learner runs in time $\mathrm{poly}(d/\epsilon)$ and outputs a halfspace with misclassification error $O(\mathrm{opt}) + \epsilon$, where $\mathrm{opt}$ is the 0-1 error of the best fitting halfspace. At a technical level, our algorithm employs an iterative soft localization technique enhanced with appropriate testers to ensure that the data distribution is sufficiently similar to a Gaussian. Finally, our algorithm can be readily adapted to yield an efficient and testable active learner requiring only $d\,\mathrm{polylog}(1/\epsilon)$ labeled examples.

## 1 Introduction

A (homogeneous) halfspace is a Boolean function $h : \mathbb{R}^d \to \{\pm 1\}$ of the form $h_{\mathbf{w}}(\mathbf{x}) = \mathrm{sign}\,(\mathbf{w} \cdot \mathbf{x})$, where $\mathbf{w} \in \mathbb{R}^d$ is the corresponding weight vector and the function $\mathrm{sign} : \mathbb{R} \to \{\pm 1\}$ is defined as $\mathrm{sign}(t) = 1$ if $t \geq 0$ and $\mathrm{sign}(t) = -1$ otherwise. Learning halfspaces from random labeled examples is a classical task in machine learning, with history going back to the Perceptron algorithm [Ros58]. In the realizable PAC model [Val84] (i.e., with consistent labels), the class of halfspaces is known to be efficiently learnable without distributional assumptions. On the other hand, in the agnostic (or adversarial label noise) model [Hau92, KSS94] even *weak* learning is computationally intractable in the distribution-free setting [Dan16, DKMR22, Tie22].

These intractability results have served as a motivation for the study of agnostic learning in the distribution-specific setting, i.e., when the marginal distribution on examples is assumed to be well-behaved. In this context, a number of algorithmic results are known. The $L_1$-regression algorithm of [KKMS08] agnostically learns halfspaces within near-optimal 0-1 error of $\mathrm{opt} + \epsilon$, where $\mathrm{opt}$ is the 0-1 error of the best-fitting halfspace. The running time of the $L_1$-regression algorithm is $d^{\tilde{O}(1/\epsilon^2)}$ under the assumption that the marginal distribution on examples is the standard Gaussian (and for a few other structured distributions) [DGJ$^+$10, DKN10]. While the $L_1$ regression method leads to improper learners, a proper agnostic learner with qualitatively similar complexity was recently given

in [DKK+21]. The exponential dependence on $1/\epsilon$ in the running time of these algorithms is known to be inherent, in both the Statistical Query model [DKZ20, GGK20, DKPZ21] and under standard cryptographic assumptions [DKR23].

Interestingly, it is possible to circumvent the super-polynomial dependence on $1/\epsilon$ by relaxing the final error guarantee — namely, by obtaining a hypothesis with 0-1 error $f(\text{opt})+\epsilon$, for some function $f(t)$ that goes to 0 when $t \to 0$. (Vanilla agnostic learning corresponds to the case that $f(t) = t$.) A number of algorithmic works, starting with [KLS09], developed efficient algorithms with relaxed error guarantees; see, e.g., [ABL17, Dan15, DKS18, DKTZ20b]. The most relevant results in the context of the current paper are the works [ABL17, DKS18] which gave $\text{poly}(d/\epsilon)$ time algorithms with error $C\text{opt} + \epsilon$, for some universal constant $C > 1$, for learning halfspaces with adversarial label noise under the Gaussian distribution. Given the aforementioned computational hardness results, these constant-factor approximations are best possible within the class of polynomial-time algorithms.

A drawback of distribution-specific agnostic learning is that it provides no guarantees if the assumption on the marginal distribution on examples is not satisfied. Ideally, one would additionally like an efficient method to *test* these distributional assumptions, so that: (1) if our tester accepts, then we can trust the output of the learner, and (2) it is unlikely that the tester rejects if the data satisfies the distributional assumptions. This state-of-affairs motivated the definition of a new model — introduced in [RV23] and termed *testable learning* — formally defined below:

**Definition 1.1** (Testable Learning with Adversarial Label Noise [RV23])**.** *Fix $\epsilon, \tau \in (0, 1]$ and let $f : [0, 1] \mapsto \mathbb{R}_+$. A tester-learner $\mathcal{A}$ (approximately) testably learns a concept class $\mathcal{C}$ with respect to the distribution $D_\mathbf{x}$ on $\mathbb{R}^d$ with $N$ samples, and failure probability $\tau$ if the following holds. For any distribution $D$ on $\mathbb{R}^d \times \{\pm 1\}$, the tester-learner $\mathcal{A}$ draws a set $S$ of $N$ i.i.d. samples from $D$. In the end, it either rejects $S$ or accepts $S$ and produces a hypothesis $h : \mathbb{R}^d \mapsto \{\pm 1\}$. Moreover, the following conditions must be met:*

- *(Completeness) If $D$ truly has marginal $D_\mathbf{x}$, $\mathcal{A}$ accepts with probability at least $1 - \tau$.*

- *(Soundness) The probability that $\mathcal{A}$ accepts and outputs a hypothesis $h$ for which $\mathbf{Pr}_{(\mathbf{x},y)\sim D}[h(\mathbf{x}) \neq y] > f(\text{opt}) + \epsilon$, where $\text{opt} := \min_{g \in \mathcal{C}} \mathbf{Pr}_{(\mathbf{x},y)\sim D}[g(\mathbf{x}) \neq y]$ is at most $\tau$.*

*The probability in the above statements is over the randomness of the sample $S$ and the internal randomness of the tester-learner $\mathcal{A}$.*

The initial work [RV23] and the followup paper [GKK22] focused on the setting where $f(t) = t$ (i.e., achieving optimal error of $\text{opt} + \epsilon$). These works developed general moment-matching based algorithms that yield testable learners for a range of concept classes, including halfspaces. For the class of halfspaces in particular, they gave a testable agnostic learner under the Gaussian distribution with sample complexity and runtime $d^{\tilde{O}(1/\epsilon^2)}$ — essentially matching the complexity of the problem in the standard agnostic PAC setting (without the testable requirement). Since the testable learning setting is at least as hard as the standard PAC setting, the aforementioned hardness results imply that the exponential complexity dependence in $1/\epsilon$ cannot be improved.

In this work, we continue this line of investigation. We ask whether we can obtain *fully polynomial time* testable learning algorithms with relaxed error guarantees — ideally matching the standard (non-testable) learning setting. Concretely, we study the following question:

> *Is there a $\text{poly}(d/\epsilon)$ time tester-learner for halfspaces with error $f(\text{opt}) + \epsilon$?*
> *Specifically, is there a constant-factor approximation?*

As our main result, we provide an affirmative answer to this question in the strongest possible sense — by providing an efficient constant-factor approximate tester-learner.

Labeling examples often requires hiring expert annotators, paying for query access to powerful language models, etc. On the other hand unlabeled examples are usually easy to obtain in practice. This has motivated the design of *active* learning algorithms that, given a large dataset of unlabeled examples, carefully choose a small subset to ask for their labels. A long line of research has studied active learning of halfspaces under structured distributions either with clean labels or in the presence of noise [BBL06, BBZ07, BL13, ABL17, She21].

In principle, testable learners could only rely on unlabeled examples during their "testing phase" in order to verify that the $\mathbf{x}$-marginal satisfies the required properties; and, assuming that it does so,

proceed to run an active learning algorithm. However, as the testing process is often coupled with the downstream learning task — that depends on labeled examples — it is an interesting question to investigate the design of testable learners in the active learning setting.

**Definition 1.2** (Testable Active Learning). *A tester-learner has sample complexity $N$ and label complexity $q$ in the active learning model if it draws $N$ unlabeled samples from $D_{\mathbf{x}}$, i.e., the $\mathbf{x}$-marginal of $D$ over $\mathbb{R}^d$, and queries the labels of at most $q$ samples throughout its computation.*

**Main Result** Our main result is the first polynomial-time tester-learner for homogeneous halfspaces with respect to the Gaussian distribution in the presence of adversarial label noise. Formally, we establish the following theorem:

**Theorem 1.3** (Testable Active Learning Halfspaces under Gaussian Marginals). *Let $\epsilon, \tau \in (0, 1)$ and $\mathcal{C}$ be the class of homogeneous halfspaces on $\mathbb{R}^d$. There exists a tester-learner with sample complexity $N = \mathrm{poly}(d, 1/\epsilon) \log(1/\tau)$ and runtime $\mathrm{poly}(d\ N)$ for $\mathcal{C}$ with respect to $\mathcal{N}(\mathbf{0}, \mathbf{I})$ up to 0-1 error $O(\mathrm{opt}) + \epsilon$, where $\mathrm{opt}$ is the 0-1 error of the best fitting function in $\mathcal{C}$, that fails with probability at most $\tau$. In addition, in the active learning model, the algorithm has sample complexity $N$ and label complexity $q = \widetilde{O}((d\ \log(1/\epsilon) + \log^2(1/\epsilon)) \log(1/\tau))$.*

Before we provide an overview of our technical approach, some remarks are in order. Theorem 1.3 gives the first algorithm for testable learning of halfspaces that runs in $\mathrm{poly}(d/\epsilon)$ time and achieves dimension-independent error (i.e., error of the form $f(\mathrm{opt}) + \epsilon$, where $f$ satisfies $\lim_{t \to 0} f(t) = 0$.) Moreover, the constant-factor approximation achieved is best possible, matching the known guarantees without the testable requirement and complexity lower bounds. Prior to our work, the only known result in the testable setting, due to [RV23, GKK22], achieves error $\mathrm{opt} + \epsilon$ with complexity $d^{\mathrm{poly}(1/\epsilon)}$. A novel (and seemingly necessary) feature of our approach is that the testing components of our algorithm depend on the labels (as opposed to the label-oblivious testers of [RV23, GKK22]). As will be explained in the proceeding discussion, to prove Theorem 1.3 we develop a testable version of the well-known localization technique that may be of broader interest.

**Independent Work** In concurrent and independent work, [GKSV23] gave an efficient tester-learner for homogeneous halfspaces under the Gaussian distribution (and strongly log-concave distributions) achieving dimension-independent error guarantees. Specifically, for strongly log-concave distributions, their algorithm achieves error $O(k^{1/2}\mathrm{opt}^{1-1/k})$ with sample complexity and running time of $\mathrm{poly}(d^{\widetilde{O}(k)}, (1/\epsilon)^{\widetilde{O}(k)})$. That is, they obtain error $O(\mathrm{opt}^c)$, where $c < 1$ is a universal constant, in $\mathrm{poly}_c(d/\epsilon)$ time; and error $\widetilde{O}(\mathrm{opt})$ in quasi-polynomial $(d/\epsilon)^{\mathrm{polylog}(d)}$ time. For the Gaussian distribution, they achieve error $O(\mathrm{opt})$ with complexity $\mathrm{poly}(d, 1/\epsilon)$. We remark that since the learner in [GKSV23] corresponds to the stochastic gradient descent algorithm from [DKTZ20b], which requires labeled sample in every iteration, it is not clear whether their approach can be made label-efficient.

## 1.1 Overview of Techniques

Our tester-learner is based on the well-known localization technique that has been used in the context of learning halfspaces with noise; see, e.g., [ABL17, DKS18]. At a high-level, the idea of localization hinges on updating a given hypothesis by using "the most informative" examples, specifically examples that have very small margin with respect to the current hypothesis. Naturally, the correctness of this geometric technique leverages structural properties of the underlying distribution over examples, namely concentration, anti-concentration, and anti-anti-concentration properties (see, e.g., [DKTZ20a]). While the Gaussian distribution satisfies these properties, they are unfortunately hard to test. In this work, we show that localization can be effectively combined with appropriate efficient testing routines to provide an efficient tester-learner.

**Localization and a (Weak) *Proper* Testable Learner** Assume that we are given a halfspace defined by the unit vector $\mathbf{w}$ with small 0-1 error, namely $\mathbf{Pr}_{\mathbf{x} \sim D_{\mathbf{x}}}[\mathrm{sign}(\mathbf{v}^* \cdot \mathbf{x}) \neq \mathrm{sign}(\mathbf{w} \cdot \mathbf{x})] \leq \delta$, for some small $\delta > 0$, where $\mathbf{v}^*$ is the unit vector defining an optimal halfspace. The localization approach improves the current hypothesis, defined by $\mathbf{w}$, by considering the conditional distribution $D'$ on the points that fall in a thin slice around $\mathbf{w}$, i.e., the set of points $\mathbf{x}$ satisfying $|\mathbf{w} \cdot \mathbf{x}| \leq O(\delta)$. The goal is to compute a new (unit) weight vector $\mathbf{w}'$ that is close to an optimal halfspace, defined by $\mathbf{v}^*$, with

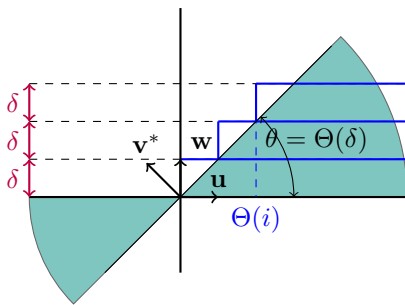

Figure 1: The disagreement region between a halfspace with normal vector $\mathbf{w}$ and the target $\mathbf{v}^*$ is shown in green. The unit direction $\mathbf{u}$ corresponds to the projection of $\mathbf{v}^*$ on the orthogonal complement of $\mathbf{w}$. We assume that the $\ell_2$ distance of the two halfspaces is $\delta$ (and thus their angle is $\Theta(\delta)$). Since the slabs $S_i = \{i\delta \leq |\mathbf{x} \cdot \mathbf{w}| \leq (i+1)\delta\}$ have width $\delta$, the $x$-coordinate of the start of the $i$-th box is $\Theta(i)$.

respect to $D'$, i.e., $\mathbf{Pr}_{\mathbf{x}' \sim D'_{\mathbf{x}}}[\mathrm{sign}(\mathbf{v}^* \cdot \mathbf{x}') \neq \mathrm{sign}(\mathbf{w}' \cdot \mathbf{x}')] \leq \alpha$, for an appropriate $\alpha > 0$. We can then show that the halfspace defined by $\mathbf{w}'$ will be *closer* to the target halfspace (defined by $\mathbf{v}^*$) with respect to the *original* distribution, i.e., we have that $\mathbf{Pr}_{\mathbf{x} \sim D_{\mathbf{x}}}[\mathrm{sign}(\mathbf{v}^* \cdot \mathbf{x}) \neq \mathrm{sign}(\mathbf{w}' \cdot \mathbf{x})] \leq O(\delta\alpha)$. By repeating the above step, we iteratively reduce the disagreement with $\mathbf{v}^*$ until we reach our target error of $O(\mathrm{opt})$. Similarly to [DKS18], instead of "hard" conditioning on a thin slice, we perform a "soft" localization step where (by rejection sampling) we transform the $\mathbf{x}$-marginal to a Gaussian whose covariance is $O(\delta^2)$ in the direction of $\mathbf{w}$ and identity in the orthogonal directions, i.e., $\mathbf{\Sigma} = \mathbf{I} - (1 - \delta^2)\mathbf{w}\mathbf{w}^\top$; see Fact 3.2.

A crucial ingredient of our approach is a ***proper* testable, weak agnostic learner** with respect to the Gaussian distribution. More precisely, our tester-learner runs in polynomial time and either reports that the $\mathbf{x}$-marginal is not $\mathcal{N}(\mathbf{0}, \mathbf{I})$ or outputs a unit vector $\mathbf{w}$ with small constant distance to the target $\mathbf{v}^*$, i.e., $\|\mathbf{w} - \mathbf{v}^*\|_2 \leq 1/100$; see Proposition 2.1. Our weak proper tester-learner first verifies that the given $\mathbf{x}$-marginal approximately matches constantly many low-degree moments with the standard Gaussian; and if it does, it returns the vector defined by the degree-1 Chow parameters, i.e., $\mathbf{c} = \mathbf{E}_{(\mathbf{x},y) \sim D}[y\mathbf{x}]$. Our main structural result in this context shows that if $D_{\mathbf{x}}$ approximately matches its low-degree moments with the standard Gaussian, then the Chow parameters of any homogeneous LTF with respect to $D_{\mathbf{x}}$ are close to its Chow parameters with respect to $\mathcal{N}(\mathbf{0}, \mathbf{I})$, i.e., for any homogeneous LTF $f(\mathbf{x})$, we have that $\mathbf{E}_{\mathbf{x} \sim D_{\mathbf{x}}}[f(\mathbf{x})\mathbf{x}] \approx \mathbf{E}_{\mathbf{x}^* \sim \mathcal{N}(\mathbf{0},\mathbf{I})}[f(\mathbf{x}^*)\mathbf{x}^*]$; see Lemma 2.3. Since the Chow vector of a homogeneous LTF with respect to the Gaussian distribution is parallel to its normal vector $\mathbf{v}^*$ (see Fact 2.2), it is not hard to show that the Chow vector of the LTF with respect to $D_{\mathbf{x}}$ will not be very far from $\mathbf{v}^*$ and will satisfy the (weak) learning guarantee of $\|\mathbf{c} - \mathbf{v}^*\|_2 \leq 1/100$. Finally, to deal with label noise, we show that if $\mathbf{x}'$ has bounded second moments (a condition that we can efficiently test), we can robustly estimate $\mathbf{E}_{\mathbf{x} \sim D_{\mathbf{x}}}[f(\mathbf{x})\mathbf{x}]$ with samples from $D$ up to error $O(\sqrt{\mathrm{opt}})$ (see Lemma 2.4), which suffices for our purpose of weak learning. The detailed description of our weak, proper tester-learner can be found in Section 2.

**From Parameter Distance to Zero-One Error** Having a (weak) testable proper learner, we can now use it on the localized (conditional) distribution $D'$ and obtain a vector $\mathbf{w}'$ that is closer to $\mathbf{v}^*$ in $\ell_2$ distance; see Lemma 3.3. However, our goal is to obtain a vector that has small zero-one disagreement with the target halfspace $\mathbf{v}^*$. Assuming that the underlying $\mathbf{x}$-marginal is a standard normal distribution, and that $\|\mathbf{w} - \mathbf{v}^*\|_2 = \delta$, it holds that $\mathbf{Pr}_{\mathbf{x} \sim D_{\mathbf{x}}}[\mathrm{sign}(\mathbf{w} \cdot \mathbf{x}) \neq \mathrm{sign}(\mathbf{v}^* \cdot \mathbf{x})] = O(\delta)$, which implies that achieving $\ell_2$-distance $O(\mathrm{opt}) + \epsilon$ suffices. We give an algorithm that can efficiently either certify that small $\ell_2$-distance implies small zero-one disagreement with respect to the given marginal $D_{\mathbf{x}}$ or declare that $D_{\mathbf{x}}$ is not the standard normal.

The disagreement region of $\mathbf{v}^*$ and $\mathbf{w}$ is a union of two "wedges" (intersection of two halfspaces); see Figure 1. In order for our algorithm to work, we need to verify that these wedges do not contain too much probability mass. Similarly to our approach for the weak tester-leaner, one could try a moment-matching approach and argue that if $D_{\mathbf{x}}$ matches its "low"-degree moments with $\mathcal{N}(\mathbf{0}, \mathbf{I})$, then small $\ell_2$-distance translates to small zero-one disagreement. However, we will need to use this result for vectors that are very close to the target (but still not close enough), namely $\|\mathbf{w} - \mathbf{v}^*\|_2 = \delta$,

where $\delta = \Theta(\epsilon)$; this would require matching $\text{poly}(1/\delta)$ many moments (as we essentially need to approximate the wedge of Figure 1 with a polynomial) and would thus lead to an exponential runtime of $d^{\text{poly}(1/\delta)}$.

Instead of trying to approximate the disagreement region with a polynomial, we will make use of the fact that our algorithm knows $\mathbf{w}$ (but not $\mathbf{v}^*$) and approximate the disagreement region by a union of cylindrical slabs. We consider slabs of the form $S_i = \{\mathbf{x} : i\delta \leq |\mathbf{w} \cdot \mathbf{x}| \leq (i+1)\delta\}$. If the target distribution is Gaussian, we know that the set $|\mathbf{w} \cdot \mathbf{x}| \gg \sqrt{\log(1/\epsilon)}$ has mass $O(\delta)$ and we can essentially ignore it. Therefore, we can cover the whole space by considering roughly $M = O(\sqrt{\log(1/\delta)}/\delta)$ slabs of width $\delta$ and split the disagreement region into the disagreement region inside each slab $S_i$. We have that

$$\Pr_{\mathbf{x} \sim D_{\mathbf{x}}} [\text{sign}(\mathbf{w} \cdot \mathbf{x}) \neq \text{sign}(\mathbf{v}^* \cdot \mathbf{x})] \leq \sum_{i=1}^{M} \Pr[|\mathbf{u} \cdot \mathbf{x}| \geq i \mid \mathbf{x} \in S_i] \ \Pr[S_i] ,$$

where $\mathbf{u}$ is the unit direction parallel to the projection of the target $\mathbf{v}^*$ onto the orthogonal complement of $\mathbf{w}$, see Figure 1. By the anti-concentration of the Gaussian distribution we know that each slab should have mass at most $O(\delta)$. Note that this is easy to test by sampling and computing empirical estimates of the mass of each slab. Moreover, assuming that underlying distribution is $\mathcal{N}(\mathbf{0}, \mathbf{I})$, we have that, conditional on $S_i$ the orthogonal direction $\mathbf{u} \cdot \mathbf{x} \sim \mathcal{N}(0, 1)$ (see Figure 1) and in particular $\mathbf{u} \cdot \mathbf{x}$ has bounded second moment. We do not know the orthogonal direction $\mathbf{u}$ as it depends on the unknown $\mathbf{v}^*$ but we can check that, conditional on the slab $S_i$, the projection of $D_{\mathbf{x}}$ onto the orthogonal complement of $\mathbf{w}$ is (approximately) mean-zero and has bounded covariance (i.e., bounded above by $2\mathbf{I}$). Note that both these conditions hold when $\mathbf{x} \sim \mathcal{N}(\mathbf{0}, \mathbf{I})$ and can be efficiently tested with samples in time $\text{poly}(d, 1/\delta)$. Under those conditions we have that that $\Pr[S_i] = O(\delta)$ for all $i$. Moreover, when the conditional distribution on $S_i$ (projected on the orthogonal complement of $\mathbf{w}$) has bounded second moment, we have that $\Pr[|\mathbf{u} \cdot \mathbf{x}| \geq i \mid \mathbf{x} \in S_i] \leq O(1/i^2)$. Combining the above, we obtain that under those assumptions the total probability of disagreement is at most $O(\delta)$. The detailed analysis is given in Section 3.1.

**Obtaining an Active Learner.** Our localization-based algorithm can be readily adapted to yield an efficient active learner-tester. We first observe that the tester that verifies that the parameter distance is proportional to 0/1 error does not require any labels, as it relies on verifying moments of the $\mathbf{x}$-marginal of the underlying distribution. During each iteration of localization, since the learner is only trying to find a constant approximation of the defining vector of the unknown halfspace, the learner only needs roughly $O(d)$ samples that fall in the slice we localize to. Since we can selectively query only the labels of samples that fall in the right region and the total number of iterations is at most $O(\log(1/\epsilon))$, the labels needed to learn the defining vectors is roughly $O(d \log(1/\epsilon))$. In the end, the learner needs to find the best halfspace among the ones from $O(\log(1/\epsilon))$ localization iterations. We remark this can also be done with $\text{polylog}(1/\epsilon)$ many label queries (see Lemma 3.6).

## 1.2 Preliminaries

We use small boldface characters for vectors and capital bold characters for matrices. We use $[d]$ to denote the set $\{1, 2, \ldots, d\}$. For a vector $\mathbf{x} \in \mathbb{R}^d$ and $i \in [d]$, $\mathbf{x}_i$ denotes the $i$-th coordinate of $\mathbf{x}$, and $\|\mathbf{x}\|_2 := \sqrt{\sum_{i=1}^{d} \mathbf{x}_i^2}$ the $\ell_2$ norm of $\mathbf{x}$. We use $\mathbf{x} \cdot \mathbf{y} := \sum_{i=1}^{n} \mathbf{x}_i \mathbf{y}_i$ as the inner product between them. We use $\mathbb{1}\{E\}$ to denote the indicator function of some event $E$.

We use $\mathbf{E}_{\mathbf{x} \sim D}[\mathbf{x}]$ for the expectation of the random variable $\mathbf{x}$ according to the distribution $D$ and $\Pr[E]$ for the probability of event $E$. For simplicity of notation, we may omit the distribution when it is clear from the context. For $\mu \in \mathbb{R}^d, \mathbf{\Sigma} \in \mathbb{R}^{d \times d}$, we denote by $\mathcal{N}(\mu, \mathbf{\Sigma})$ the $d$-dimensional Gaussian distribution with mean $\mu$ and covariance $\mathbf{\Sigma}$. For $(\mathbf{x}, y) \in \mathcal{X}$ distributed according to $D$, we denote $D_{\mathbf{x}}$ to be the marginal distribution of $\mathbf{x}$. Let $f : \mathbb{R}^d \mapsto \{\pm 1\}$ be a boolean function and $D$ a distribution over $\mathbb{R}^d$. The degree-1 Chow parameter vector of $f$ with respect to $D$ is defined as $\mathbf{E}_{\mathbf{x} \sim D}[f(\mathbf{x})\mathbf{x}]$. For a halfspace $h(\mathbf{x}) = \text{sign}(\mathbf{v} \cdot \mathbf{x})$, we say that $\mathbf{v}$ is the defining vector of $h$.

**Moment-Matching** In what follows, we use the phrase *"A distribution $D$ on $\mathbb{R}^d$ matches $k$ moments with a distribution $Q$ up to error $\Delta$"*. Similarly to [GKK22], we formally define approximate moment-matching as follows.

**Definition 1.4** (Approximate Moment-Matching). *Let $k \in \mathbb{N}$ be a degree parameter and let $\mathcal{M}(k, d)$ be the set of $d$-variate monomials of degree up to $k$. Moreover, let $\boldsymbol{\Delta} \in \mathbb{R}_+^{|\mathcal{M}(k,d)|}$ be a slack parameter (indexed by the monomials of $\mathcal{M}(k, d)$), satisfying $\boldsymbol{\Delta}_0 = 0$. We say that two distributions $D, Q$ match $k$ moments up to error $\boldsymbol{\Delta}$ if $|\mathbf{E}_{\mathbf{x} \sim D}[m(\mathbf{x})] - \mathbf{E}_{\mathbf{x} \sim Q}[m(\mathbf{x})]| \leq \boldsymbol{\Delta}_m$ for every monomial $m(\mathbf{x}) \in \mathcal{M}(k, d)$. When the error bound $\Delta$ is the same for all monomials we overload notation and simply use $\Delta$ instead of the parameter $\boldsymbol{\Delta}$.*

## 2 Weak Testable Proper Agnostic Learning

As our starting point, we give an algorithm that performs testable *proper* learning of homogeneous halfspaces in the presence of adversarial label noise with respect to the Gaussian distribution. The main result of this section is given below and its proof can be found in Appendix A.

**Proposition 2.1** (Proper Testable Learner with Adversarial Label Noise). *Let $D$ be a distribution on labeled examples $(\mathbf{x}, y) \in \mathbb{R}^d \times \{\pm 1\}$ with $\mathbf{x}$-marginal $D_{\mathbf{x}}$. Suppose that there exists a unit vector $\mathbf{v}^* \in \mathbb{R}^d$ such that $\mathbf{Pr}_{(\mathbf{x},y) \sim D}\left[\mathrm{sign}(\mathbf{v}^* \cdot \mathbf{x}) \neq y\right] \leq \mathrm{opt}$. There exists an algorithm (Algorithm 2) that given $\tau, \eta \in (0, 1)$, $N = d^{\widetilde{O}(1/\eta^2)} \log(1/\tau)$ i.i.d. unlabeled samples from $D_{\mathbf{x}}$, queries the labels of $O(d/\eta^2) \log(d/\tau)$ of them, and runs in time $\mathrm{poly}(d, N)$ and does one of the following:*

- *The algorithm reports that the $\mathbf{x}$-marginal of $D$ is not $\mathcal{N}(\mathbf{0}, \mathbf{I})$.*

- *The algorithm outputs a unit vector $\mathbf{w} \in \mathbb{R}^d$.*

*With probability at least $1 - \tau$ the following holds: (1) if the algorithm reports anything, the report is correct, and (2) if the algorithm returns a vector $\mathbf{w}$, it holds $\|\mathbf{v}^* - \mathbf{w}\|_2 \leq C_{\mathrm{A}}\sqrt{\mathrm{opt} + \eta}$, where $C_{\mathrm{A}} > 0$ is an absolute constant.*

A couple of remarks are in order. First, notice that if the algorithm outputs a vector $\mathbf{w}$, we only have the guarantee that $\|\mathbf{v}^* - \mathbf{w}\|_2$ is small — instead of that the hypothesis halfspace $h_{\mathbf{w}}(\mathbf{x}) = \mathrm{sign}(\mathbf{w} \cdot \mathbf{x})$ achieves small 0-1 error. Nonetheless, as we will show in the next section, conditioned on $D$ passing some test, the error of the halfspace $h_{\mathbf{w}}$ will be at most $\mathrm{opt}$ plus a constant multiple of $\|\mathbf{v}^* - \mathbf{w}\|_2$ (see Lemma 3.1). Second, unlike the testable *improper* learners in [RV23, GKK22] — which achieve error of $\mathrm{opt} + \eta$ with similar running time and sample complexity — our testable *proper* learner achieves the weaker error guarantee of $O(\sqrt{\mathrm{opt} + \eta})$. This suffices for our purposes for the following reason: in the context of our localization-based approach, we only need an efficient proper *weak* learner that achieves sufficiently small *constant* error. This holds for our proper testable learner, as long as both $\mathrm{opt}$ and $\eta$ are bounded above by some other sufficiently small constant.

To obtain a proper learner, we proceed to directly estimate the defining vector $\mathbf{v}^*$ of the target halfspace $h^*(\mathbf{x}) = \mathrm{sign}(\mathbf{v}^* \cdot \mathbf{x})$, where we assume without loss of generality that $\mathbf{v}^*$ is a unit vector. The following simple fact relating the degree-1 *Chow-parameters* of a homogeneous halfspace and its defining vector will be useful for us.

**Fact 2.2** (see, e.g., Lemma 4.3 of [DKS18]). *Let $\mathbf{v}$ be a unit vector and $h(\mathbf{x}) = \mathrm{sign}(\mathbf{v} \cdot \mathbf{x})$ be the corresponding halfspace. If $\mathbf{x}$ is from $\mathcal{N}(\mathbf{0}, \mathbf{I})$, then we have that $\mathbf{E}_{\mathbf{x} \sim \mathcal{N}(\mathbf{0},\mathbf{I})}\left[h(\mathbf{x})\mathbf{x}\right] = \sqrt{2/\pi}\,\mathbf{v}$.*

To apply Fact 2.2 in our context, we need to overcome two hurdles: (i) the $\mathbf{x}$ marginal of $D$ is not necessarily the standard Gaussian, and (ii) the labels are not always consistent with $h^*(\mathbf{x})$. The second issue can be circumvented by following the approach of [DKS18]. In particular, if the $\mathbf{x}$ marginal of $D$ is indeed Gaussian, we can just treat $D$ as a corrupted version of $(\mathbf{x}, h^*(\mathbf{x}))$, where $\mathbf{x} \sim \mathcal{N}(\mathbf{0}, \mathbf{I})$ and estimate the Chow parameters robustly.

To deal with the first issue, we borrow tools from [GKK22]. At a high level, we certify that the low-degree moments of $D_{\mathbf{x}}$ — the $\mathbf{x}$ marginal of $D$ — approximately match the corresponding moments of $\mathcal{N}(\mathbf{0}, \mathbf{I})$ before estimating the Chow parameters. Importantly, this testing procedure uses only unlabeled samples. To establish the correctness of our algorithm, we show that, for any distribution $B$ that passes the moment test, the Chow parameters of a halfspace under $B$ will still be close to its defining vector. We state the lemma formally below. Its proof can be found in Appendix A.

**Lemma 2.3** (From Moment-Matching to Chow Distance). *Fix $\eta > 0$. Let $k = C\log(1/\eta)/\eta^2$ and $\Delta = \frac{1}{kd^k}\left(\frac{1}{C\sqrt{k}}\right)^{k+1}$, where $C > 0$ is a sufficiently large absolute constant. Let $B$ be a*

*distribution whose moments up to degree $k$ match with those of $\mathcal{N}(\mathbf{0}, \mathbf{I})$ up to additive error $\Delta$. Let $h(\mathbf{x}) = \text{sign}(\mathbf{v} \cdot \mathbf{x})$ be a halfspace. Then we have that $\left\| \mathbf{E}_{\mathbf{x} \sim B}[h(\mathbf{x})\mathbf{x}] - \sqrt{\frac{2}{\pi}}\,\mathbf{v} \right\|_2 \leq O(\sqrt{\eta})$ .*

With Lemma 2.3 in hand, we know it suffices to estimate the Chow parameters of $h^*$ with respect to $D_{\mathbf{x}}$. This would then give us a good approximation to $\mathbf{v}^*$ conditioned on $D_{\mathbf{x}}$ indeed having its low-degree moments approximately match those of $\mathcal{N}(\mathbf{0}, \mathbf{I})$. We use the following lemma, which estimates the Chow parameters *robustly* under adversarial label noise.

**Lemma 2.4.** *Let $\eta, \tau \in (0, 1)$ and $G$ be a distribution over $\mathbb{R}^d \times \{\pm 1\}$ such that $\mathbf{E}_{\mathbf{x} \sim G_{\mathbf{x}}}[\mathbf{x}\mathbf{x}^\top] \preccurlyeq 2\mathbf{I}$. Let $\mathbf{v} \in \mathbb{R}^d$ be a unit vector such that $\mathbf{Pr}_{(\mathbf{x},y) \sim G}[\text{sign}(\mathbf{v} \cdot \mathbf{x}) \neq y] \leq \eta$. Then there exists an algorithm that takes $N = O(d/\eta^2) \log(d/\tau)$ labeled samples from $G$, runs in time $\text{poly}(N)$, and outputs a vector $\mathbf{w}$ such that $\|\mathbf{E}_{\mathbf{x} \sim G_{\mathbf{x}}}[\text{sign}(\mathbf{v} \cdot \mathbf{x})\mathbf{x}] - \mathbf{w}\|_2 \leq O(\sqrt{\eta})$ with probability at least $1 - \tau$.*

The proof follows from standard arguments in robust mean estimation and the details can be found in Appendix C. We remark that the number of labeled samples required by this procedure is (nearly) linear with respect to $d$ for constant $\eta$, which is important for us to obtain (nearly) optimal label complexity overall.

We are now ready to describe our testable proper learner. In particular, it first certifies that the moments of the underlying distribution match with the standard Gaussian and then draws i.i.d. samples to estimate the Chow parameter. It then follows from Fact 2.2, Lemmas 2.3 and 2.4 that the resulting Chow vector, conditioned on that the moment test pass, is closed to the defining vector of the optimal halfspace. We given the pseudo-code of the algorith and its analysis, which also serves as the proof of Propostion 2.1, in the end of Appendix A.

# 3 Efficient Testable Learning of Halfspaces

In this section, we give our tester-learner for homogeneous halfspaces under the Gaussian distribution, thereby proving Theorem 1.3. Throughout this section, we will fix an optimal halfspace $h^*(\mathbf{x}) = \text{sign}(\mathbf{v}^* \cdot \mathbf{x})$, i.e., a halfspace with optimal 0-1 error.

The structure of this section is as follows: In Section 3.1, we present a tester which certifies that the probability of the disagreement region between two halfspaces whose defining vectors are close to each other is small under $D_{\mathbf{x}}$. In Section 3.2, we present and analyze our localization step and combine it with the tester from Section 3.1 to obtain our final algorithm.

## 3.1 From Parameter Distance to 0-1 Error

For two homogeneous halfspaces $h_{\mathbf{u}}(\mathbf{x}) = \text{sign}(\mathbf{u} \cdot \mathbf{x})$ and $h_{\mathbf{v}}(\mathbf{x}) = \text{sign}(\mathbf{v} \cdot \mathbf{x})$, where $\mathbf{u}, \mathbf{v}$ are unit vectors, if $D_{\mathbf{x}}$ is the standard Gaussian, $\mathcal{N}(\mathbf{0}, \mathbf{I})$, we can express the probability mass of their disagreement region as follows (see, e.g., Lemma 4.2 of [DKS18]):

$$\Pr_{\mathbf{x} \sim D_{\mathbf{x}}}[h_{\mathbf{u}}(\mathbf{x}) \neq h_{\mathbf{v}}(\mathbf{x})] \leq O\left(\|\mathbf{u} - \mathbf{v}\|_2\right) . \tag{1}$$

Hence, learning homogeneous halfspaces under Gaussian marginals can often be reduced to approximately learning the defining vector of some optimal halfspace $h^*$. This is no longer the case if $D_{\mathbf{x}}$ is an arbitrary distribution, which may well happen in our regime. We show in this section that it is still possible to "certify" whether some relationship similar to the one in Equation (1) holds.

In particular, given a known vector $\mathbf{v}$, we want to make sure that for any other vector $\mathbf{w}$ that is close to $\mathbf{v}$, the mass of the disagreement region between the halfspaces defined by by $\mathbf{v}, \mathbf{w}$ respectively is small. To do so, we will decompose the space into many thin "slabs" that are stacked on top of each other in the direction of $\mathbf{v}$. Then, we will certify the mass of disagreement restricted to each of the slab is not too large. For slabs that are close to the halfspace $\text{sign}(\mathbf{v} \cdot \mathbf{x})$, we can check these slabs must not themselves be too heavy. For slabs that are far away from the halfspace, we use the observation that the points in the disagreement region must then have large components in the subspace perpendicular to $\mathbf{v}$. Hence, as long as $D$ has its second moment bounded, we can bound the mass of the disagreement region in these far-away slabs using standard concentration inequality. Formally, we have the following lemma, whose proof can be found in Appendix B.

**Lemma 3.1** (Wedge Bound). *Let $D_{\mathbf{x}}$ be a distribution over $\mathbb{R}^d$. Given a unit vector $\mathbf{v}$ and parameters $\eta, \tau \in (0, 1/2)$, there exists an algorithm (Algorithm 1) that draws i.i.d. samples from $D_{\mathbf{x}}$, runs in*

---

**Input:** Sample access to a distribution $D_{\mathbf{x}}$ over $\mathbb{R}^d$; tolerance parameter $\eta > 0$; unit vector $\mathbf{v} \in \mathbb{R}^d$; failure probability $\tau \in (0, 1)$.
**Output:** Certifies that for all unit vectors $\mathbf{w}$ such that $\|\mathbf{w} - \mathbf{v}\|_2 \leq \eta$ it holds that $\mathbf{Pr}_{\mathbf{x} \sim D_{\mathbf{x}}}[\text{sign}(\mathbf{v} \cdot \mathbf{x}) \neq \text{sign}(\mathbf{w} \cdot \mathbf{x})] \leq C\eta$, for some absolute constant $C > 1$, or reports that $D_{\mathbf{x}}$ is not $\mathcal{N}(\mathbf{0}, \mathbf{I})$.

1. Set $B = \lceil \sqrt{\log(1/\eta)}/\eta \rceil$.

2. Let $\widetilde{D}$ be the empirical distribution obtained by drawing $\text{poly}(d, 1/\eta) \log(1/\tau)$ samples from $D_{\mathbf{x}}$.

3. For integers $-B - 1 \leq i \leq B$, define $E_i$ to be the event that $\{\mathbf{v} \cdot \mathbf{x} \in [i\eta, (i+1)\eta]\}$ and $E_{B+1}$ to be the event that $\{|\mathbf{v} \cdot \mathbf{x}| \geq \sqrt{\log(1/\eta)}\}$.

4. Verify that $\sum_{i=-B-1}^{B+1} \left| \mathbf{Pr}_{\mathcal{N}(\mathbf{0},\mathbf{I})}[E_i] - \mathbf{Pr}_{\widetilde{D}}[E_i] \right| \leq \eta$.

5. Let $S_i$ the distribution of $\widetilde{D}$ conditioned on $E_i$ and $S_i^\perp$ be $S_i$ projected on the subspace orthogonal to $\mathbf{v}$.

6. For each $i$, verify that $S_i^\perp$ has bounded covariance, i.e., check that $\mathbf{E}_{\mathbf{x} \sim S_i^\perp}[\mathbf{x}\mathbf{x}^\top] \preccurlyeq 2\mathbf{I}$.

---

**Algorithm 1:** Wedge-Bound

time $\text{poly}(d, 1/\eta) \log(1/\tau)$, and reports either one of the following: (i) For all unit vectors $\mathbf{w}$ such that $\|\mathbf{w} - \mathbf{v}\|_2 \leq \eta$ it holds $\mathbf{Pr}_{\mathbf{x} \sim D_{\mathbf{x}}}[\text{sign}(\mathbf{v} \cdot \mathbf{x}) \neq \text{sign}(\mathbf{w} \cdot \mathbf{x})] \leq C\eta$, for some absolute constant $C > 1$. (ii) $D_{\mathbf{x}}$ is not the standard Gaussian $\mathcal{N}(\mathbf{0}, \mathbf{I})$. Moreover, with probability at least $1 - \tau$, the report is accurate.

## 3.2    Algorithm and Analysis: Proof of Theorem 1.3

We employ the idea of "soft" localization used in [DKS18]. In particular, given a vector $\mathbf{v}$ and a parameter $\sigma$, we use rejection sampling to define a new distribution $D_{\mathbf{v},\sigma}$ that "focuses" on the region near the halfspace $\text{sign}(\mathbf{v} \cdot \mathbf{x})$.

**Fact 3.2** (Rejection Sampling, Lemma 4.7 of [DKS18])**.** *Let $D$ be a distribution on labeled examples* $(\mathbf{x}, y) \in \mathbb{R}^d \times \{\pm 1\}$. *Let $\mathbf{v} \in \mathbb{R}^d$ be a unit vector and $\sigma \in (0, 1)$. We define the distribution $D_{\mathbf{v},\sigma}$ as follows: draw a sample $(\mathbf{x}, y)$ from $D$ and accept it with probability $e^{-(\mathbf{v} \cdot \mathbf{x})^2 \cdot (\sigma^{-2}-1)/2}$. Then, $D_{\mathbf{v},\sigma}$ is the distribution of $(\mathbf{x}, y)$ conditional on acceptance. If the $\mathbf{x}$-marginal of $D$ is $\mathcal{N}(\mathbf{0}, \mathbf{I})$, then the $\mathbf{x}$-marginals of $D_{\mathbf{v},\sigma}$ is $\mathcal{N}(\mathbf{0}, \mathbf{\Sigma})$, where $\mathbf{\Sigma} = \mathbf{I} - (1 - \sigma^2)\mathbf{v}\mathbf{v}^T$. Moreover, the acceptance probability of a point is $\sigma$.*

We now briefly discuss the sample and label complexities of this rejection sampling procedure. Since the acceptance probability is $\sigma$, we need roughly $1/\sigma$ samples from $D$ to simulate one sample from $D_{\mathbf{v},\sigma}$. On the other hand, to simulate one *labeled* sample from $D$, one only need 1 label query in addition to the unlabeled samples as one can query only the points that are accepted by the procedure. This makes sure that rejection sampling is efficient in terms of its label complexity.

The main idea of localization is the following. Let $\mathbf{v}$ be a vector such that $\|\mathbf{v} - \mathbf{v}^*\|_2 \leq \delta$. Suppose that we use localization to the distribution $D_{\mathbf{v},\delta}$. If we can learn a halfspace with defining vector $\mathbf{w}$ that achieves sufficiently small constant error with respect to the new distribution $D_{\mathbf{v},\delta}$, we can then combine our knowledge of $\mathbf{w}$ and $\mathbf{v}$ to produce a new halfspace with significantly improved error guarantees under $D$. The following lemma formalizes this geometric intuition.

**Lemma 3.3.** *Let $\mathbf{v}^*, \mathbf{v}$ be two unit vectors in $\mathbb{R}^d$ such that $\|\mathbf{v} - \mathbf{v}^*\|_2 \leq \delta \leq 1/100$. Let $\mathbf{\Sigma} = \mathbf{I} - (1 - \delta^2)\mathbf{v}\mathbf{v}^T$ and $\mathbf{w}$ be a unit vector such that $\left\| \mathbf{w} - \frac{\mathbf{\Sigma}^{1/2}\mathbf{v}^*}{\|\mathbf{\Sigma}^{1/2}\mathbf{v}^*\|_2} \right\|_2 \leq \zeta \leq 1/100$. Then it holds $\left\| \frac{\mathbf{\Sigma}^{-1/2}\mathbf{w}}{\|\mathbf{\Sigma}^{-1/2}\mathbf{w}\|_2} - \mathbf{v}^* \right\|_2 \leq 5(\delta^2 + \delta\zeta)$.*

We defer the proof to Appendix C. Below, we provide a few useful remarks regarding the relevant parameters.

**Remark 3.4.** Observe that in Lemma 3.3 we require that the distance of $\mathbf{v}$ and $\mathbf{v}^*$ is smaller than $1/100$. While this constant is not the best possible, we remark that some non-trivial error is indeed necessary so that the localization step works. For example, assume that $\mathbf{v}$ and $\mathbf{v}^*$ are orthogonal, i.e., $\mathbf{v} \cdot \mathbf{v}^* = 0$, and that $\mathbf{\Sigma} = \mathbf{I} - (1 - \xi^2)\mathbf{v}\mathbf{v}^T$ for some $\xi \in [0, 1]$. Observe that $\mathbf{\Sigma}^{1/2}$ scales vectors by a factor of $\xi$ in the direction of $\mathbf{v}$ and leaves orthogonal directions unchanged. Similarly, its inverse $\mathbf{\Sigma}^{-1/2}$ scales vectors by a factor of $1/\xi$ in the direction of $\mathbf{v}$ and leaves orthogonal directions unchanged. Without loss of generality, assume that $\mathbf{v} = \mathbf{e}_1, \mathbf{v}^* = \mathbf{e}_2$ where $\mathbf{e}_1, \mathbf{e}_2$ are the two standard basis vectors. Then $\mathbf{\Sigma}^{1/2}\mathbf{v}^* = \mathbf{v}^*$. Moreover, assume that $\mathbf{w} = a\mathbf{e}_1 + b\mathbf{e}_2$ (with $a^2 + b^2 = 1$). We observe that $\|\mathbf{w} - \mathbf{\Sigma}^{1/2}\mathbf{v}^*/\|\mathbf{\Sigma}^{1/2}\mathbf{v}^*\|_2\|_2^2 = \|\mathbf{w} - \mathbf{v}^*\|_2^2 = 2 - 2b$. However, we have that $\mathbf{s} = \mathbf{\Sigma}^{-1/2}\mathbf{w} = (a/\xi)\mathbf{e}_1 + b\mathbf{e}_2$. Therefore, $\|\mathbf{s}/\|\mathbf{s}\|_2 - \mathbf{v}^*\|_2^2 = 2 - 2b/\sqrt{(a/\xi)^2 + b^2}$. Notice that for all $\xi \in [0, 1]$ it holds that $\mathbf{s}/\|\mathbf{s}\|_2$ is further away from $\mathbf{v}^*$ than $\mathbf{w}$, i.e., rescaling by $\mathbf{\Sigma}^{-1/2}$ worsens the error.

We are now ready to present an iterative testing/learning procedure that will serve as the main component of our algorithm. At a high level, we first use Algorithm 2 from Proposition 2.1 to learn a vector $\mathbf{v}$ that is close to $\mathbf{v}^*$ in $\ell_2$-distance up to some sufficiently small constant. Then we localize to the learned halfspace and re-apply Algorithm 2 to iteratively improve $\mathbf{v}$. In particular, we will argue that, whenever the learned halfspace is still significantly suboptimal, the algorithm either detects that the underlying distribution is not Gaussian or keeps making improvements such that $\mathbf{v}$ gets closer to $\mathbf{v}^*$ conditioned on $\mathbf{v}$ still being sub-optimal. The pseudocode and analysis of the algorithm can be found in Appendix C.

**Lemma 3.5.** *Suppose* $\left\|\mathbf{v}^* - \mathbf{v}^{(t)}\right\|_2 \leq \delta \leq 1/100$. *There is an algorithm (Algorithm 3) that with probability at least* $1 - \tau$, *either (i) correctly reports that the* $\mathbf{x}$*-marginal of $D$ is not* $\mathcal{N}(\mathbf{0}, \mathbf{I})$ *or (ii) computes a unit vector* $\mathbf{v}^{(t+1)}$ *so that either* $\left\|\mathbf{v}^* - \mathbf{v}^{(t+1)}\right\|_2 \leq \delta/2$ *or* $\left\|\mathbf{v}^* - \mathbf{v}^{(t)}\right\|_2 \leq C\mathrm{opt}$, *where $C > 0$ is an absolute constant. Furthermore, the algorithm draws* $N = \mathrm{poly}(d, 1/\delta) \log(1/\tau)$ *many unlabeled samples and uses* $O\left(d \, \log(d/\tau)\right)$ *label queries.*

After running Algorithm 3 for at most a logarithmic number of iterations, we know $\mathbf{v}$ must be close to $\mathbf{v}^*$. Then, we can use Algorithm 1 from Lemma 3.1 to certify that the disagreement between the learned halfspace and $h^*(\mathbf{x})$ is small. One technical difficulty is that we do not know precisely when to terminate the update and each update does not necessarily monotonically bring $\mathbf{v}$ closer to $\mathbf{v}^*$. As a result, the process only yields a list of vectors $\mathbf{v}^{(1)}, \cdots \mathbf{v}^{(t)}$ with the guarantee that at least one of them is sufficiently closed to $\mathbf{v}^*$. A natural idea is to estimate the 0-1 errors of the halfspaces defined by the vectors in the list and simply pick the one with the smallest empirical error. Naively, we need to estimate the errors up to an additive $\epsilon$ and this may take up to $\Omega(1/\epsilon^2)$ labeled samples, which is exponentially worse than our goal in terms of the dependency on $\epsilon$. Nonetheless, we remark that comparing the errors of two halfspaces which differ by some multiplicative factors can be done much more efficiently in terms of its label complexity. In particular, it suffices for us to compare the errors of the two halfspaces *restricted* to the area in which the predictions made by them differ. Under this conditioning, the difference between the errors then gets magnified to some constant, making comparisons much easier. Hence, we can run a tournament among the the halfspaces, which reduces to perform pairwise comparisons among the lists, to obtain our final winner hypothesis. The result is summarized in the following lemma.

**Lemma 3.6** (Tournament). *Let $\epsilon, \tau \in (0, 1)$ and $D$ a distribution over $\mathbb{R}^d \times \{\pm 1\}$. Given a list of halfspaces $\{h^{(i)}\}_{i=1}^k$, there is an algorithm that draws* $N = \Theta\left(k^2 \log(k/\tau)/\epsilon\right)$ *i.i.d. unlabeled samples from $D_{\mathbf{x}}$, uses* $\Theta\left(k^2 \log(k/\tau)\right)$ *label queries, runs in time* $\mathrm{poly}(d, N)$ *and outputs a halfspace $\tilde{h}$ satisfying that* $\mathbf{Pr}_{(\mathbf{x},y)\sim D}[\tilde{h}(x) \neq y] \leq 10 \, \min_i \mathbf{Pr}_{(\mathbf{x},y)\sim D}[h^{(i)}(x) \neq y] + \epsilon$.

We provide the pseudocode of the algorithm and its analysis in Appendix C. The proof of Theorem 1.3 then follows.

Our final algorithm (Algorithm 5) simply iteratively applies the localized update routine (Algorithm 3) and then runs a tournament (Algorithm 4) among the $O(\log(1/\epsilon))$ candidate halfspaces obtained from the iterations. The pseudocode and its analysis, which serves as proof of our main theorem, can be found at the end of Appendix C.

# 4 Acknowledgements

ID was supported by NSF Medium Award CCF-2107079, NSF Award CCF-1652862 (CAREER), and a DARPA Learning with Less Labels (LwLL) grant. DK was supported by NSF Medium Award CCF-2107547 and NSF Award CCF-1553288 (CAREER). NZ was supported in part by NSF award 2023239, NSF Medium Award CCF-2107079, and a DARPA Learning with Less Labels (LwLL) grant.

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

# Supplementary Material

## A    Omitted Proofs for Weak Testable Proper Agnostic Learning

*Proof of Lemma 2.3.* It suffices to show that for any unit vector $\mathbf{u} \in \mathbb{R}^d$, the following holds:

$$\left| \mathop{\mathbf{E}}_{\mathbf{x} \sim B} [h(\mathbf{x})\mathbf{x} \cdot \mathbf{u}] - \mathop{\mathbf{E}}_{\mathbf{x} \sim \mathcal{N}(\mathbf{0}, \mathbf{I})} [h(\mathbf{x})\mathbf{x} \cdot \mathbf{u}] \right| \leq O(\sqrt{\eta}) \ .$$

The following fact expresses a real number $a$ as an integral of the sign function.

**Fact A.1.** *For any $a \in \mathbb{R}$, it holds $a = \frac{1}{2} \int_0^\infty (\mathrm{sign}(a - t) + \mathrm{sign}(a + t))\mathrm{d}t$.*

We apply Fact A.1 to the term $\mathbf{u} \cdot \mathbf{x}$, which gives

$$\left| \mathop{\mathbf{E}}_{\mathbf{x} \sim B} [h(\mathbf{x})\mathbf{x} \cdot \mathbf{u}] - \mathop{\mathbf{E}}_{\mathbf{x} \sim \mathcal{N}(\mathbf{0}, \mathbf{I})} [h(\mathbf{x})\mathbf{x} \cdot \mathbf{u}] \right| = \frac{1}{2} \left| \mathop{\mathbf{E}}_{\mathbf{x} \sim B} \left[ h(\mathbf{x}) \int_{t \geq 0} (\mathrm{sign}(\mathbf{u} \cdot \mathbf{x} - t) + \mathrm{sign}(\mathbf{u} \cdot \mathbf{x} + t)) \, \mathrm{d}t \right] \right.$$

$$\left. - \mathop{\mathbf{E}}_{\mathbf{x} \sim \mathcal{N}(\mathbf{0}, \mathbf{I})} \left[ h(\mathbf{x}) \int_{t \geq 0} (\mathrm{sign}(\mathbf{u} \cdot \mathbf{x} - t) + \mathrm{sign}(\mathbf{u} \cdot \mathbf{x} + t)) \, \mathrm{d}t \right] \right|$$

$$= \frac{1}{2} \left| \int_{t \geq 0} \left( \mathop{\mathbf{E}}_{\mathbf{x} \sim B} [h(\mathbf{x}) (\mathrm{sign}(\mathbf{u} \cdot \mathbf{x} - t) + \mathrm{sign}(\mathbf{u} \cdot \mathbf{x} + t))] \right. \right.$$

$$\left. \left. - \mathop{\mathbf{E}}_{\mathbf{x} \sim \mathcal{N}(\mathbf{0}, \mathbf{I})} [h(\mathbf{x}) (\mathrm{sign}(\mathbf{u} \cdot \mathbf{x} - t) + \mathrm{sign}(\mathbf{u} \cdot \mathbf{x} + t))] \right) \mathrm{d}t \right| ,$$

where in the last line we switch the order of the integral of $t$ and $\mathbf{x}$ by Fubini's theorem. We then split the above integral over $t$ into two parts based on the magnitude of $t$ ($t > 1/\sqrt{\eta}$ versus $0 \leq t \leq 1/\sqrt{\eta}$) and apply the triangle inequality:

$$\left| \mathop{\mathbf{E}}_{\mathbf{x} \sim B} [h(\mathbf{x})\mathbf{x} \cdot \mathbf{u}] - \mathop{\mathbf{E}}_{\mathbf{x} \sim \mathcal{N}(\mathbf{0}, \mathbf{I})} [h(\mathbf{x})\mathbf{x} \cdot \mathbf{u}] \right|$$

$$\leq \frac{1}{2} \left| \int_{0 \leq t \leq 1/\sqrt{\eta}} \left( \mathop{\mathbf{E}}_{\mathbf{x} \sim B} [h(\mathbf{x})\mathrm{sign}(\mathbf{u} \cdot \mathbf{x} - t)] - \mathop{\mathbf{E}}_{\mathbf{x} \sim \mathcal{N}(\mathbf{0}, \mathbf{I})} [h(\mathbf{x})\mathrm{sign}(\mathbf{u} \cdot \mathbf{x} - t)] \right) \mathrm{d}t \right|$$

$$+ \frac{1}{2} \left| \int_{0 \leq t \leq 1/\sqrt{\eta}} \left( \mathop{\mathbf{E}}_{\mathbf{x} \sim B} [h(\mathbf{x})\mathrm{sign}(\mathbf{u} \cdot \mathbf{x} + t)] - \mathop{\mathbf{E}}_{\mathbf{x} \sim \mathcal{N}(\mathbf{0}, \mathbf{I})} [h(\mathbf{x})\mathrm{sign}(\mathbf{u} \cdot \mathbf{x} + t)] \right) \mathrm{d}t \right|$$

$$+ \frac{1}{2} \left| \int_{t \geq 1/\sqrt{\eta}} \left( \mathop{\mathbf{E}}_{\mathbf{x} \sim B} [h(\mathbf{x}) (\mathrm{sign}(\mathbf{u} \cdot \mathbf{x} - t) + \mathrm{sign}(\mathbf{u} \cdot \mathbf{x} + t))] \right. \right.$$

$$\left. \left. - \mathop{\mathbf{E}}_{\mathbf{x} \sim \mathcal{N}(\mathbf{0}, \mathbf{I})} [h(\mathbf{x}) (\mathrm{sign}(\mathbf{u} \cdot \mathbf{x} - t) + \mathrm{sign}(\mathbf{u} \cdot \mathbf{x} + t))] \right) \mathrm{d}t \right| . \tag{2}$$

We start by bounding the integral for $t \geq 1/\sqrt{\eta}$.

**Lemma A.2** (Chow-Distance Tail). *Let $Q$ be distribution over $\mathbb{R}^d$ with $\mathbf{E}_{\mathbf{x} \sim Q}[\mathbf{x}\mathbf{x}^\top] \preccurlyeq 2\mathbf{I}$. Moreover, let $g(\mathbf{x}) : \mathbb{R}^d \mapsto \mathbb{R}$ be a bounded function, i.e., $|g(\mathbf{x})| \leq 1$ for all $\mathbf{x} \in \mathbb{R}^d$. It holds*

$$\left| \int_{t \geq 1/\sqrt{\eta}} \mathop{\mathbf{E}}_{\mathbf{x} \sim Q} [g(\mathbf{x}) (\mathrm{sign}(\mathbf{u} \cdot \mathbf{x} - t) + \mathrm{sign}(\mathbf{u} \cdot \mathbf{x} + t))] \, \mathrm{d}t \right| \leq O(\sqrt{\eta}) .$$

*Proof.* We split the expectation into two parts based on the relative sizes of $|\mathbf{u} \cdot \mathbf{x}|$ and $t$. Specifically, we can write:

$$\left| \int_{t \geq 1/\sqrt{\eta}} \mathop{\mathbf{E}}_{\mathbf{x} \sim Q} \left[ g(\mathbf{x}) \left( \text{sign}(\mathbf{u} \cdot \mathbf{x} - t) + \text{sign}(\mathbf{u} \cdot \mathbf{x} + t) \right) \right] \mathrm{d}t \right|$$

$$\leq \left| \int_{t \geq 1/\sqrt{\eta}} \mathop{\mathbf{E}}_{\mathbf{x} \sim Q} \left[ g(\mathbf{x}) \left( \text{sign}(\mathbf{u} \cdot \mathbf{x} - t) + \text{sign}(\mathbf{u} \cdot \mathbf{x} + t) \right) \mathbb{1}\{|\mathbf{u} \cdot \mathbf{x}| \geq t\} \right] \mathrm{d}t \right|$$

$$+ \left| \int_{t \geq 1/\sqrt{\eta}} \mathop{\mathbf{E}}_{\mathbf{x} \sim Q} \left[ g(\mathbf{x}) \left( \text{sign}(\mathbf{u} \cdot \mathbf{x} - t) + \text{sign}(\mathbf{u} \cdot \mathbf{x} + t) \right) \mathbb{1}\{|\mathbf{u} \cdot \mathbf{x}| \leq t\} \right] \mathrm{d}t \right|. \tag{3}$$

For the second term in Equation (3), we rely on the following observation: when $|\mathbf{u} \cdot \mathbf{x}| \leq t$, the quantities $\mathbf{u} \cdot \mathbf{x} - t$ and $\mathbf{u} \cdot \mathbf{x} + t$ have opposite signs. Hence, we conclude the integrand is $0$ everywhere and therefore the second term is also $0$. For the first term, we have

$$\left| \int_{t \geq 1/\sqrt{\eta}} \mathop{\mathbf{E}}_{\mathbf{x} \sim Q} \left[ g(\mathbf{x}) \left( \text{sign}(\mathbf{u} \cdot \mathbf{x} - t) + \text{sign}(\mathbf{u} \cdot \mathbf{x} + t) \right) \mathbb{1}\{|\mathbf{u} \cdot \mathbf{x}| \geq t\} \right] \right|$$

$$\leq \int_{t \geq 1/\sqrt{\eta}} \mathop{\mathbf{E}}_{\mathbf{x} \sim Q} \left[ \left| g(\mathbf{x}) \left( \text{sign}(\mathbf{u} \cdot \mathbf{x} - t) + \text{sign}(\mathbf{u} \cdot \mathbf{x} + t) \right) \mathbb{1}\{|\mathbf{u} \cdot \mathbf{x}| \geq t\} \right| \right]$$

$$\leq \int_{t \geq 1/\sqrt{\eta}} \mathop{\mathbf{E}}_{\mathbf{x} \sim Q} \left[ 2 \mathbb{1}\{|\mathbf{u} \cdot \mathbf{x}| \geq t\} \right] \leq 4 \int_{t \geq 1/\sqrt{\eta}} \frac{1}{t^2} \leq O(\sqrt{\eta}),$$

where the first inequality follows from the triangle inequality, the second inequality uses the fact that the $\text{sign}(\cdot)$ function is at most $1$ and the third inequality follows from Chebyshev's inequality using the fact that the $\mathbf{E}[\mathbf{x}\mathbf{x}^\top] \preccurlyeq 2\mathbf{I}$. Combining our analysis for the two terms in Equation (3), we can then conclude the proof of Lemma A.2. $\qquad\square$

Using the triangle inequality and applying Lemma A.2 on the distributions $B$ and $\mathcal{N}(\mathbf{0}, \mathbf{I})$, we have that

$$\frac{1}{2} \left| \int_{t \geq 1/\sqrt{\eta}} \left( \mathop{\mathbf{E}}_{\mathbf{x} \sim B} \left[ h(\mathbf{x}) \left( \text{sign}(\mathbf{u} \cdot \mathbf{x} - t) + \text{sign}(\mathbf{u} \cdot \mathbf{x} + t) \right) \right] \right. \right.$$

$$\left. \left. - \mathop{\mathbf{E}}_{\mathbf{x} \sim \mathcal{N}(\mathbf{0}, \mathbf{I})} \left[ h(\mathbf{x}) \left( \text{sign}(\mathbf{u} \cdot \mathbf{x} - t) + \text{sign}(\mathbf{u} \cdot \mathbf{x} + t) \right) \right] \right) \mathrm{d}t \right| \leq O(\sqrt{\eta}). \tag{4}$$

We then turn our attention to the terms

$$\left| \int_{0 \leq t \leq 1/\sqrt{\eta}} \left( \mathop{\mathbf{E}}_{\mathbf{x} \sim B} \left[ h(\mathbf{x}) \text{sign}(\mathbf{u} \cdot \mathbf{x} - t) \right] - \mathop{\mathbf{E}}_{\mathbf{x} \sim \mathcal{N}(\mathbf{0}, \mathbf{I})} \left[ h(\mathbf{x}) \text{sign}(\mathbf{u} \cdot \mathbf{x} - t) \right] \right) \mathrm{d}t \right|. \tag{5}$$

$$\left| \int_{0 \leq t \leq 1/\sqrt{\eta}} \left( \mathop{\mathbf{E}}_{\mathbf{x} \sim B} \left[ h(\mathbf{x}) \text{sign}(\mathbf{u} \cdot \mathbf{x} + t) \right] - \mathop{\mathbf{E}}_{\mathbf{x} \sim \mathcal{N}(\mathbf{0}, \mathbf{I})} \left[ h(\mathbf{x}) \text{sign}(\mathbf{u} \cdot \mathbf{x} + t) \right] \right) \mathrm{d}t \right|. \tag{6}$$

To bound Equations (5) and (6), we need the following fact from [GKK22].

**Fact A.3** (Theorem 5.6 of [GKK22]). *Let $h : \mathbb{R}^d \mapsto \{\pm 1\}$ be a function of $p$ halfspaces, i.e., $h(\mathbf{x}) = g(h_1(\mathbf{x}), \cdots, h_p(\mathbf{x}))$ where $h_i$ are halfspaces and $g : \{\pm 1\}^p \mapsto \{\pm 1\}$. For any $k \in \mathbb{N}$, let $\Delta = \frac{\sqrt{p}}{2k} \frac{1}{d^k} \left( \frac{1}{C'\sqrt{k}} \right)^{k+1}$ for some sufficiently large absolute constant $C' > 0$. Then, for any distribution $B$ whose moments up to order $k$ match those of $\mathcal{N}(\mathbf{0}, \mathbf{I})$ up to $\Delta$, we have*

$$\left| \mathop{\mathbf{E}}_{\mathbf{x} \sim \mathcal{N}(\mathbf{0}, \mathbf{I})} [h(\mathbf{x})] - \mathop{\mathbf{E}}_{\mathbf{x} \sim B} [h(\mathbf{x})] \right| \leq \frac{1}{\sqrt{k}} \sqrt{p} \left( C \log \left( \sqrt{pk} \right) \right)^{2p}.$$

*for some constant $C > 0$.*

For a fixed $t$, note that $h(\mathbf{x})\mathrm{sign}(\mathbf{u} \cdot \mathbf{x} - t)$ is a function of two halfspaces. Moreover, from the assumptions of Lemma 2.3, the distributions $B$ and $\mathcal{N}(\mathbf{0}, \mathbf{I})$ match $k = C\log(1/\eta)/\eta^2$ moments up to error $\Delta = \frac{1}{kd^k}\left(\frac{1}{C\sqrt{k}}\right)^{k+1}$, where $C > 0$ is a sufficiently large absolute constant. Therefore, applying Fact A.3 and the triangle inequality gives

$$\frac{1}{2}\left|\int_{0\leq t\leq 1/\sqrt{\eta}}\left(\mathop{\mathbf{E}}_{\mathbf{x}\sim B}[h(\mathbf{x})\mathrm{sign}(\mathbf{u} \cdot \mathbf{x} - t)] - \mathop{\mathbf{E}}_{\mathbf{x}\sim\mathcal{N}(\mathbf{0},\mathbf{I})}[h(\mathbf{x})\mathrm{sign}(\mathbf{u} \cdot \mathbf{x} - t)]\right)\mathrm{d}t\right|$$
$$\leq O(1)\int_{0\leq t\leq 1/\sqrt{\eta}}\eta\mathrm{d}t = O(\sqrt{\eta}) . \tag{7}$$

Similarly, we can show that Equation (6) is bounded by $O(\sqrt{\eta})$. Substituting the bounds from Equations (4) and (7) into Equation (2) then gives

$$\left|\mathop{\mathbf{E}}_{\mathbf{x}\sim B}[h(\mathbf{x})\mathbf{x} \cdot \mathbf{u}] - \mathop{\mathbf{E}}_{\mathbf{x}\sim\mathcal{N}(\mathbf{0},\mathbf{I})}[h(\mathbf{x})\mathbf{x} \cdot \mathbf{u}]\right| \leq O(\sqrt{\eta}).$$

Since $\mathbf{u}$ is chosen as an arbitrary unit vector, this implies that

$$\left\|\mathop{\mathbf{E}}_{\mathbf{x}\sim B}[h(\mathbf{x})\mathbf{x}] - \mathop{\mathbf{E}}_{\mathbf{x}\sim\mathcal{N}(\mathbf{0},\mathbf{I})}[h(\mathbf{x})\mathbf{x}]\right\|_2 \leq O(\sqrt{\eta}).$$

Combining this with Fact 2.2 concludes the proof of Lemma 2.3. $\qquad\square$

*Proof of Lemma 2.4.* We first show that $\left\|\mathbf{E}_{\mathbf{x}\sim G_{\mathbf{x}}}[\mathrm{sign}(\mathbf{v} \cdot \mathbf{x})\,\mathbf{x}] - \mathbf{E}_{(\mathbf{x},y)\sim G}[y\,\mathbf{x}]\right\|_2 \leq O(\sqrt{\eta})$. For any unit vector $\mathbf{u}$, we have that

$$\mathop{\mathbf{E}}_{\mathbf{x}\sim G_{\mathbf{x}}}[\mathrm{sign}(\mathbf{v} \cdot \mathbf{x})\,\mathbf{u} \cdot \mathbf{x}] - \mathop{\mathbf{E}}_{(\mathbf{x},y)\sim G}[y\,\mathbf{u} \cdot \mathbf{x}] = \mathop{\mathbf{E}}_{(\mathbf{x},y)\sim G}[(\mathrm{sign}(\mathbf{v} \cdot \mathbf{x}) - y)\mathbf{u} \cdot \mathbf{x}]$$
$$\leq \sqrt{\mathop{\mathbf{E}}_{(\mathbf{x},y)\sim G}[(\mathrm{sign}(\mathbf{v} \cdot \mathbf{x}) - y)^2]\mathop{\mathbf{E}}_{\mathbf{x}\sim G_{\mathbf{x}}}[(\mathbf{u} \cdot \mathbf{x})^2]}$$
$$\leq 4\sqrt{\eta} ,$$

where we used the Cauchy-Schwarz inequality and the fact that $\mathbf{Pr}_{(\mathbf{x},y)\sim G}[\mathrm{sign}(\mathbf{v} \cdot \mathbf{x}) \neq y] \leq \eta$. Therefore, we have that $\left\|\mathbf{E}_{\mathbf{x}\sim G_{\mathbf{x}}}[\mathrm{sign}(\mathbf{v} \cdot \mathbf{x})\mathbf{x}] - \mathbf{E}_{(\mathbf{x},y)\sim G}[y\mathbf{x}]\right\|_2 \leq 4\sqrt{\eta}$. Let $(\mathbf{x}^{(1)}, y^{(1)}), \ldots, (\mathbf{x}^{(N_1)}, y^{(N_1)})$ be samples drawn from $G$, where $N_1 = O(d/\eta^2)$. Then, let $\widetilde{\mathbf{w}} = (1/N_1)\sum_{j=1}^{N_1} y^{(j)}\mathbf{x}^{(j)}$. From Chebyshev's inequality, we know the $i$-th coordinate of $\widetilde{\mathbf{w}}$ should satisfy $\mathbf{Pr}[|\widetilde{w}_i - \mathbf{E}_{(\mathbf{x},y)\sim D}[y\mathbf{x}]| \geq \eta/\sqrt{d}] \leq 4d/(N_1\eta^2) \leq 1/4$. Therefore, using the standard median technique, we can find a $\mathbf{w}_i^{\mathrm{median}}$, so that $\mathbf{Pr}[|\mathbf{w}_i^{\mathrm{median}} - \mathbf{E}_{(\mathbf{x},y)\sim D}[y\mathbf{x}]| \geq \eta/\sqrt{d}] \leq \tau/d$, using $N_2 = O(N_1\log(d/\tau))$ samples. Let $\mathbf{w} = (\mathbf{w}_1^{\mathrm{median}}, \ldots, \mathbf{w}_d^{\mathrm{median}})$, then we have that $\left\|\mathbf{E}_{(\mathbf{x},y)\sim G}[y\mathbf{x}] - \mathbf{w}\right\|_2 \leq O(\eta)$ with probability at least $1 - \tau$. Then, using the triangle inequality, we have that $\|\mathbf{E}_{\mathbf{x}\sim G_{\mathbf{x}}}[\mathrm{sign}(\mathbf{v} \cdot \mathbf{x})\mathbf{x}] - \mathbf{w}\|_2 \leq O(\sqrt{\eta})$, which concludes the proof of Lemma 2.4. $\qquad\square$

*Proof of Proposition 2.1.* Let $k, \Delta, N$ be defined as in Algorithm 2. If $D_{\mathbf{x}}$ is $\mathcal{N}(\mathbf{0}, \mathbf{I})$, the moments up to degree $k$ of the $\mathbf{x}$-marginal of the empirical distribution $\widehat{D}_N$ (obtained after drawing $N$ i.i.d. samples from $D_{\mathbf{x}}$) are close to those of $\mathcal{N}(\mathbf{0}, \mathbf{I})$ up to additive error $\Delta$ with probability at least $1 - \tau/10$.

If Algorithm 2 did not terminate on Line 3, we then have that the moments up to degree $k$ of $\widehat{D}_N$ are close to those of $\mathcal{N}(\mathbf{0}, \mathbf{I})$ up to additive error $\Delta$ with probability at least $1 - \tau/10$. Then, applying Lemma 2.3 with $B = \widehat{D}_N$ and $h(\mathbf{x}) = \mathrm{sign}(\mathbf{v}^* \cdot \mathbf{x})$, we get that

$$\left\|\mathop{\mathbf{E}}_{\mathbf{x}\sim\widehat{D}_N}[\mathrm{sign}(\mathbf{v}^* \cdot \mathbf{x})\mathbf{x}] - \sqrt{2/\pi}\mathbf{v}^*\right\|_2 \leq O(\sqrt{\eta}). \tag{8}$$

Now, suppose we were to query the labels of all points in $\widehat{D}_N$. We could then construct another empirical distribution $\widehat{L}_N$ over labeled samples. By our assumption, the error of $\mathrm{sign}(\mathbf{v}^* \cdot \mathbf{x})$ under

**Input:** Sample access to a distribution $D_{\mathbf{x}}$ over unlabeled samples and query access to the labels of the drawn samples according to $D$; certification range $\eta$; failure probability $\tau$.
**Output:** Either reports that the $D_{\mathbf{x}}$ is not $\mathcal{N}(\mathbf{0}, \mathbf{I})$; or returns a unit vector $\mathbf{w} \in \mathbb{R}^d$ such that $\|\mathbf{v}^* - \mathbf{w}\|_2 \leq C_A \sqrt{\mathrm{opt} + \eta}$.

1. Set $k = C \log(1/\eta)/\eta^2$ and $\Delta = \frac{1}{kd^k} \left( \frac{1}{C\sqrt{k}} \right)^{k+1}$, where $C > 0$ is a sufficiently large absolute constant.

2. Draw $N = d^{Ck \log k} \log(1/\tau)$ samples from $D_{\mathbf{x}}$ and construct the empirical distribution $\widehat{D}_N$.

3. Certify that the moments of $\widehat{D}_N$ up to degree $k$ match with those of $\mathcal{N}(\mathbf{0}, \mathbf{I})$ up to error $\Delta$.

4. If the above does not hold; report that $D_{\mathbf{x}}$ is not $\mathcal{N}(\mathbf{0}, \mathbf{I})$ and terminate.

5. Draw $N' = O(d/\eta^2) \log(d/\eta)$ samples from $\widehat{D}_N$ and query their labels. Denote the resulting labeled samples as $S$.

6. Use algorithm from Lemma 2.4 on $S$ and obtain $\mathbf{w}$. Return $\mathbf{w}/\|\mathbf{w}\|_2$.

**Algorithm 2:** Proper Testable Learner

$D$ is at most $\mathrm{opt}$. Hence, the error of $\mathrm{sign}(\mathbf{v}^* \cdot \mathbf{x})$ under $\widehat{L}_N$ is at most $\mathrm{opt} + \eta$ with probability at least $1 - \tau/10$. Next, notice that the set of samples $S$ in Line 5 are exactly i.i.d. samples from $\widehat{L}_N$. Hence, applying Lemma 2.4 on $\widehat{L}_N$ then gives that, with probability at least $1 - \tau/10$, the vector $\mathbf{w}$, computed on Line 6 of Algorithm 2 satisfies

$$\left\| \mathop{\mathbf{E}}_{\mathbf{x} \sim (\widehat{L}_N)_{\mathbf{x}}} [\mathrm{sign}(\mathbf{v}^* \cdot \mathbf{x})\mathbf{x}] - \mathbf{w} \right\|_2 \leq O\left( \sqrt{\mathrm{opt} + \eta} \right) . \tag{9}$$

Notice that the $\mathbf{x}$-marginal of $\widehat{L}_N$ is exactly $\widehat{D}_N$. Hence, combining Equations (8) and (9), we get that $\left\| \mathbf{w} - \sqrt{2/\pi}\mathbf{v}^* \right\|_2 \leq O\left( \sqrt{\mathrm{opt} + \eta} \right)$ , as desired. $\qquad\square$

## B  Proof of Wedge Bound Lemma 3.1

*Proof of Lemma 3.1.* Recall that $\widetilde{D}$ is the empirical distribution made up of $N$ i.i.d. samples from $D_{\mathbf{x}}$ where $N = \mathrm{poly}(d, 1/\eta) \log(1/\tau)$. We consider $\widetilde{D}$ restricted to a set of thin "slabs" stacked on each other in the direction of $\mathbf{v}$. More formally, we define $S_i$ to be the distribution of $\widetilde{D}$ conditioned on $\mathbf{v} \cdot \mathbf{x} \in [(i-1)\eta, i\eta]$, for $i \in [-\sqrt{\log(1/\eta)}/\eta, \sqrt{\log(1/\eta)}/\eta]$, and $S_i^\perp$ to be the distribution $S_i$ projected into the subspace orthogonal to $\mathbf{v}$.

Suppose that $D_{\mathbf{x}}$ is indeed $\mathcal{N}(\mathbf{0}, \mathbf{I})$. Then the distribution $\widetilde{D}$ is the empirical distribution formed by samples taken from $\mathcal{N}(\mathbf{0}, \mathbf{I})$. In this case, it is easy to see that both Line 4 and 6 of Algorithm 1 pass with high probability.

**Claim B.1.** *Assume that $D_{\mathbf{x}} = \mathcal{N}(\mathbf{0}, \mathbf{I})$. Then the tests at Line 4 and 6 of Algorithm 1 pass with probability at least $1 - \tau/10$.*

*Proof.* If $\mathbf{x} \sim \mathcal{N}(\mathbf{0}, \mathbf{I})$, then $\mathbf{v} \cdot \mathbf{x} \sim \mathcal{N}(0, 1)$. If we concatenate the values of $\mathbf{Pr}_{\mathcal{N}(\mathbf{0}, \mathbf{I})}[E_i]$ into a vector, it can be viewed as the discretization of $\mathcal{N}(0, 1)$ into $2B + 3$ many buckets. On the other hand, $\mathbf{Pr}_{\widetilde{D}}[E_i]$ is an empirical version of this discrete distribution composed of $N$ i.i.d. samples where $N = \mathrm{poly}(d, 1/\eta) \log(1/\tau)$. Since we can learn any discrete distribution with support $n$ up to error $\eta$ in total variation distance with $\Theta(n/\eta^2) \log(1/\tau)$ samples with probability at least $1 - \tau$, it follows that Line 4 will pass with high probability as long as we take more than $\Theta(B/\eta^2) \log(1/\tau) \leq \mathrm{poly}(d, 1/\eta) \log(1/\tau)$ many samples.

For Line 6, we remark that $S_i^\perp$ is the empirical version of a $(d-1)$-dimensional standard Gaussian. Since the empirical mean and the empirical covariance concentrates around the true mean and

covariance with probability at least $1 - \tau$ if one takes more than $\Theta(d^2/\eta^2) \log(1/\tau)$ many samples, it follows that Line 4 will pass with high probability as long as we take more than $\Theta(d^2/\eta^2) \log(1/\tau) \leq \text{poly}(d, 1/\eta) \log(1/\tau)$ many samples. □

Suppose that both lines pass. We claim that this implies the following: for all unit vectors $\mathbf{w}$ such that $\|\mathbf{w} - \mathbf{v}\|_2 \leq \eta$ it holds

$$\Pr_{\mathbf{x} \sim \widetilde{D}} [\text{sign}(\mathbf{v} \cdot \mathbf{x}) \neq \text{sign}(\mathbf{w} \cdot \mathbf{x})] \leq C\eta \,. \tag{10}$$

for some absolute constant $C > 1$. Given this, we can deduce that the same equation must also hold for $D_\mathbf{x}$ with high probability — albeit with a larger constant $C'$. To see this, we remark that the left hand side of the equation can be treated as the error of the halfspace $\text{sign}(\mathbf{w} \cdot \mathbf{x})$ if the true labels are generated by $\text{sign}(\mathbf{v} \cdot \mathbf{x})$. Since the VC-dimension of the class of homogeneous halfspaces is $d$, we have that the error for all $\mathbf{w}$ under $D_\mathbf{x}$ is well-approximated by that under $\widetilde{D}$ up to an additive $\eta$ with probability at least $1 - \tau$ given $N = \text{poly}(d, 1/\eta) \log(1/\tau)$ many samples. Hence, conditioned on Equation (10), it holds with probability at least $1 - \tau$ that

$$\Pr_{\mathbf{x} \sim D_\mathbf{x}} [\text{sign}(\mathbf{v} \cdot \mathbf{x}) \neq \text{sign}(\mathbf{w} \cdot \mathbf{x})] \leq (C + 1)\eta \,,$$

for all $\mathbf{w}$.

We now proceed to show Equation (10) holds if the Algorithm 1 did not terminate on Lines 4 and 6. Conditioned on Line 6, for any unit vector $\mathbf{u} \in \mathbb{R}^d$ that is orthogonal to $\mathbf{v}$, we have $|\mathbf{E}[\mathbf{u} \cdot \mathbf{x}]| \leq O(1)$ and $\text{Var}[\mathbf{u} \cdot \mathbf{x}] \leq 2$. Using Chebyshev's inequality, for any $\alpha > 0$, it holds

$$\Pr_{\mathbf{x} \sim S_i^\perp} [|\mathbf{u} \cdot \mathbf{x}| \geq \alpha] \leq O \left( \frac{1 + \eta^2}{\alpha^2} \right) \,. \tag{11}$$

We can now bound $\Pr_{\mathbf{x} \sim \widetilde{D}} [\text{sign}(\mathbf{v} \cdot \mathbf{x}) \neq \text{sign}(\mathbf{w} \cdot \mathbf{x})]$ for an arbitrary unit vector $\mathbf{w}$ satisfying $\|\mathbf{w} - \mathbf{v}\|_2 \leq \eta$. We proceed to rewrite $\mathbf{w}$ as $(1 - \gamma^2)^{1/2}\mathbf{v} + \gamma\mathbf{u}$ for some unit vector $\mathbf{u} \in \mathbb{R}^d$ that is orthogonal to $\mathbf{v}$ and $\gamma \in (0, \eta)$. Denote $\gamma' = \gamma/(1 - \gamma^2)^{1/2}$, then the event that $\text{sign}(\mathbf{v} \cdot \mathbf{x}) \neq \text{sign}(\mathbf{w} \cdot \mathbf{x})$ implies that $\gamma' |\mathbf{u} \cdot \mathbf{x}| \geq |\mathbf{v} \cdot \mathbf{x}|$. Therefore, we have that

$$\Pr_{\mathbf{x} \sim S_i} [\text{sign}(\mathbf{v} \cdot \mathbf{x}) \neq \text{sign}(\mathbf{w} \cdot \mathbf{x})] \leq \Pr_{\mathbf{x} \sim S_i} [\gamma' |\mathbf{u} \cdot \mathbf{x}| \geq |\mathbf{v} \cdot \mathbf{x}|] \leq \Pr_{\mathbf{x} \sim S_i^\perp} [\gamma' |\mathbf{u} \cdot \mathbf{x}| \geq i\eta] \leq O \left( \frac{1 + \eta^2}{i^2} \right) , \tag{12}$$

where in the second inequality we use the definition of $S_i$, and in the third inequality we use that $\gamma \leq \eta$ and Equation (11). We now bound from above the total disagreement probability between $\mathbf{w}$ and $\mathbf{v}$ under $\widetilde{D}$. We have that

$$\Pr_{\mathbf{x} \sim \widetilde{D}} [\text{sign}(\mathbf{v} \cdot \mathbf{x}) \neq \text{sign}(\mathbf{w} \cdot \mathbf{x})] \leq \Pr_{\mathbf{x} \sim \widetilde{D}} \left[ |\mathbf{v} \cdot \mathbf{x}| \geq \sqrt{\log(1/\eta)} \right] + \Pr_{\mathbf{x} \sim \widetilde{D}} [|\mathbf{v} \cdot \mathbf{x}| \leq \eta]$$

$$+ \sum_{|i| > 1}^{\sqrt{\log(1/\eta)}/\eta} \Pr_{\mathbf{x} \sim S_i} [\text{sign}(\mathbf{v} \cdot \mathbf{x}) \neq \text{sign}(\mathbf{w} \cdot \mathbf{x})] \Pr_{\mathbf{x} \sim \widetilde{D}} [(i-1)\eta \leq \mathbf{x} \cdot \mathbf{v} \leq i\eta]$$

$$\leq 6\eta + O(\eta) \sum_{|i| > 1}^{\sqrt{\log(1/\eta)}/\eta} \frac{1 + \eta^2}{i^2} \leq O(\eta) \,,$$

where we used that $\Pr_{\mathbf{x} \sim \widetilde{D}}[|\mathbf{x} \cdot \mathbf{v}| > \sqrt{\log(1/\eta)}] \leq 3\eta$ and $\Pr_{\mathbf{x} \sim \widetilde{D}}[|\mathbf{x} \cdot \mathbf{v}| < \eta] \leq 3\eta$, since in Line 4 we verified that the probabilities $\Pr_{\mathbf{x} \sim \widetilde{D}}[\mathbf{v} \cdot \mathbf{x} \in [(i-1)\eta, i\eta]]$ are close to the probabilities under $\mathcal{N}(\mathbf{0}, \mathbf{I})$ and hence bounded by $O(\eta)$, Equation (12), and the fact that the series $\sum_i \frac{1}{i^2}$ is convergent and less than $\pi^2/6$. □

## C  Omitted Proofs of localization and main theorem

*Proof of Lemma 3.3.* Since $\|\mathbf{v} - \mathbf{v}^*\|_2 \leq \delta$, we can write

$$\mathbf{v}^* = \frac{1}{\sqrt{1 + \kappa^2}} (\mathbf{v} + \kappa\mathbf{u}) \tag{13}$$

for some $\kappa \in [0, \delta]$ and some unit vector $\mathbf{u}$ perpendicular to $\mathbf{v}$. By definition, $\mathbf{\Sigma}^{1/2}$ shrinks in the direction of $\mathbf{v}$ by a factor of $\delta$ and leaves other orthogonal directions unchanged. Hence, it holds

$$\mathbf{\Sigma}^{1/2}\mathbf{v}^* = \frac{1}{\sqrt{1 + \kappa^2}} \left(\delta\mathbf{v} + \kappa\mathbf{u}\right).$$

Then, using the triangle inequality, we obtain

$$\gamma := \left\|\mathbf{\Sigma}^{1/2}\mathbf{v}^*\right\|_2 \leq \frac{1}{\sqrt{1 + \kappa^2}} \left(\delta \left\|\mathbf{v}\right\|_2 + \kappa \left\|\mathbf{u}\right\|_2\right) \leq \frac{2\delta}{\sqrt{1 + \kappa^2}} \leq 2\delta,$$

since $\kappa$ is upper bounded by $\delta$. Since $\left\|\mathbf{w} - \frac{\mathbf{\Sigma}^{1/2}\mathbf{v}^*}{\left\|\mathbf{\Sigma}^{1/2}\mathbf{v}^*\right\|_2}\right\|_2 \leq \zeta$, we can write

$$\mathbf{w} = \frac{\mathbf{\Sigma}^{1/2}\mathbf{v}^*}{\left\|\mathbf{\Sigma}^{1/2}\mathbf{v}^*\right\|_2} + a\mathbf{v} + b\mathbf{u}',$$

for some $|a|, b \in [0, \zeta]$ and $\mathbf{u}'$ perpendicular to $\mathbf{v}$. We can multiply both sides by $\gamma$ and get

$$\gamma\mathbf{w} = \mathbf{\Sigma}^{1/2}\mathbf{v}^* + a\gamma\mathbf{v} + b\gamma\mathbf{u}',$$

which implies that

$$\gamma\mathbf{\Sigma}^{-1/2}\mathbf{w} = \mathbf{v}^* + a\gamma/\delta\mathbf{v} + b\gamma\mathbf{u}'.$$

Using Equation (13), we then have

$$\gamma\mathbf{\Sigma}^{-1/2}\mathbf{w} = \left(\frac{1}{\sqrt{1 + \kappa^2}} + a\frac{\gamma}{\delta}\right)\mathbf{v} + \frac{\kappa}{\sqrt{1 + \kappa^2}}\mathbf{u} + b\gamma\mathbf{u}'. \tag{14}$$

Let $\lambda := \frac{1}{\sqrt{1+\kappa^2}} + a\gamma/\delta$ be the coefficient before $\mathbf{v}$. We next identify the range of $\lambda$.

**Claim C.1.** *It holds that $\lambda \in [1 - \delta - 2\zeta, 1 + 2\zeta]$.*

*Proof.* Recall that $\kappa \in [0, \delta]$, $|a| \in [0, \zeta]$ and $\gamma \in [0, 2\delta]$. If we view $\lambda$ as a function of $\kappa, a, \gamma$, it is minimized when $\kappa = \delta, a = -\zeta, \gamma = 2\delta$, which then gives

$$\lambda \geq \frac{1}{\sqrt{1 + \delta^2}} - 2\zeta \geq 1 - \delta - 2\zeta,$$

where in the second inequality we use the fact that $\frac{1}{\sqrt{1+x^2}} \geq 1 - x$ for $x \geq 0$. On the other hand, $\lambda$ is maximized when $\kappa = 0, a = \zeta, \gamma = 2\delta$, which gives $\lambda \leq 1 + 2\zeta$. Hence, we can conclude that $\lambda \in [1 - \delta - 2\zeta, 1 + 2\zeta]$. $\qquad\square$

We multiply both sides of Equation (14) by $\frac{1}{\lambda\sqrt{1+\kappa^2}}$, which gives

$$\frac{\gamma}{\lambda\sqrt{1+\kappa^2}}\mathbf{\Sigma}^{-1/2}\mathbf{w} = \frac{1}{\sqrt{1+\kappa^2}}\mathbf{v} + \frac{\kappa}{\lambda(1+\kappa^2)}\mathbf{u} + b\frac{\kappa}{\lambda\sqrt{1+\kappa^2}}\mathbf{u}'$$

$$= \mathbf{v}^* + \left(\frac{\kappa}{\lambda(1+\kappa^2)} - \frac{\kappa}{\sqrt{1+\kappa^2}}\right)\mathbf{u} + b\frac{\gamma}{\lambda\sqrt{1+\kappa^2}}\mathbf{u}',$$

where in the second equality we use Equation (13). We then bound from above and below the norm, and we get that

$$\left|\left\|\frac{\gamma}{\lambda\sqrt{1+\kappa^2}}\mathbf{\Sigma}^{-1/2}\mathbf{w}\right\|_2 - 1\right| \leq \frac{\kappa}{1+\kappa^2}\left|\frac{1}{\lambda} - \sqrt{1+\kappa^2}\right| + \frac{b\gamma}{\lambda\sqrt{1+\kappa^2}},$$

where we used triangle inequality. Note that

$$|(1/\lambda) - \sqrt{1+\kappa^2}| \leq |1/\lambda - 1| + |1 - \sqrt{1+\kappa^2}| \leq (\delta + 2\zeta)/(1 - \delta - 2\zeta) + \kappa$$

and that $\kappa \leq \delta$. Therefore, we obtain that

$$\left|\left\|\frac{\gamma}{\lambda\sqrt{1+\kappa^2}}\mathbf{\Sigma}^{-1/2}\mathbf{w}\right\|_2 - 1\right| \leq 4(\delta^2 + \delta\zeta).$$

**Input:** Sample access to a distribution $D$ over labeled examples; unit vector $\mathbf{v} \in \mathbb{R}^d$; parameters $\tau, \eta, \delta > 0$.
**Output:** Either reports that $D_{\mathbf{x}}$ is not $\mathcal{N}(\mathbf{0}, \mathbf{I})$ or computes a unit vector $\mathbf{v}'$ such that: either $\|\mathbf{v}^* - \mathbf{v}'\|_2 \leq \delta/2$ or $\|\mathbf{v}^* - \mathbf{v}'\|_2 \leq O(\mathrm{opt})$

1. Let $D_{\mathbf{v}, \delta}$ be the distribution obtained by running Rejection Sampling with parameters $\mathbf{v}$ and $\delta$.

2. Check the acceptance probability is within $[\delta/2, 3\delta/2]$. Otherwise, report $\mathbf{x}$-marginals of $D$ is not $\mathcal{N}(\mathbf{0}, \mathbf{I})$.

3. Let $G$ be the distribution obtained by applying the transformation $\boldsymbol{\Sigma}^{-1/2}$ on the $\mathbf{x}$ marginals of $D_{\mathbf{v}, \delta}$ where $\boldsymbol{\Sigma} = \mathbf{I} - (1 - \delta^2)\mathbf{v}\mathbf{v}^\top$.

4. Run Algorithm 2 on $G$ with accuracy $\eta = 1/(20000 C_\mathrm{A}^2)$ to obtain a unit vector $\mathbf{w}$.

5. If the algorithm reports the marginal of $G$ is not $\mathcal{N}(\mathbf{0}, \mathbf{I})$, terminate and report it.

6. Set $\mathbf{v}' = \boldsymbol{\Sigma}^{1/2}\mathbf{w}/\left\|\boldsymbol{\Sigma}^{1/2}\mathbf{w}\right\|_2$ and return $\mathbf{v}'$.

**Algorithm 3:** Testable Localized-Update

Let $A = \left\|\frac{\gamma}{\lambda\sqrt{1+\kappa^2}}\boldsymbol{\Sigma}^{-1/2}\mathbf{w}\right\|_2$. We have that

$$\left\|\frac{\boldsymbol{\Sigma}^{-1/2}\mathbf{w}}{\|\boldsymbol{\Sigma}^{-1/2}\mathbf{w}\|_2} - \mathbf{v}^*\right\|_2 \leq \|\mathbf{v}^*\|_2 |1 - 1/A| + |1/A - 1| \leq 5(\delta^2 + \delta\zeta) \ .$$

This concludes the proof. $\qquad\square$

*Proof of Lemma 3.5.* For simplicity, we denote $\mathbf{v}^{(t)}$ as $\mathbf{v}$. We apply the Rejection Sampling procedure from Fact 3.2 in the direction of $\mathbf{v}$ with $\sigma = \delta$. If the $\mathbf{x}$-marginal of $D$ is $\mathcal{N}(\mathbf{0}, \mathbf{I})$, the acceptance probability of $D_{\mathbf{v}, \delta}$ is exactly $\delta$. We then estimate the acceptance probability with accuracy $\epsilon$; if it is not lying inside the interval $[\delta/2, 3\delta/2]$ (see Line 2 of Algorithm 3), we report that the $\mathbf{x}$-marginal of $D$ is not standard normal and terminate.

Conditioned on the event that the algorithm did not terminate, we can get an (unlabeled) sample from the $\mathbf{x}$-marginal of $D_{\mathbf{v}, \delta}$, using $O(1/\delta)$ unlabeled samples from $D_{\mathbf{x}}$. To get a labeled sample from $D_{\mathbf{v}, \delta}$, we only need 1 additional label query. Note that, under the distribution $D_{\mathbf{v}, \delta}$, the error of the $\mathbf{v}^*$ is

$$
\begin{aligned}
\Pr_{(\mathbf{x},y)\sim D_{\mathbf{v},\delta}}[\mathrm{sign}(\mathbf{v}^* \cdot \mathbf{x}) \neq y] &= \Pr_{(\mathbf{x},y)\sim D}[\mathrm{sign}(\mathbf{v}^* \cdot \mathbf{x}) \neq y \mid (\mathbf{x}, y) \text{ is accepted}] \\
&\leq \Pr_{(\mathbf{x},y)\sim D}[\mathrm{sign}(\mathbf{v}^* \cdot \mathbf{x}) \neq y] / \Pr_{(\mathbf{x},y)\sim D}[(\mathbf{x}, y) \text{ is accepted}] \\
&\leq 2\mathrm{opt}/\delta \ ,
\end{aligned}
\tag{15}
$$

where we used that the probability of the acceptance is at least $\delta/2$. Denote by $G$ the distribution of $(\boldsymbol{\Sigma}^{-1/2}\mathbf{x}, y)$, where $(\mathbf{x}, y) \sim D_{\mathbf{v}, \delta}$ and $\boldsymbol{\Sigma} = \mathbf{I} - (1 - \delta^2)\mathbf{v}\mathbf{v}^\top$. We note that if the $\mathbf{x}$-marginal of $D$ were the standard normal, then $\mathbf{x}$-marginal of $G$ is the standard normal. Hence, we can apply the algorithm Algorithm 2 from Proposition 2.1. Under the transformed distribution, we have that the new optimal vector $(\mathbf{v}^*)' := \boldsymbol{\Sigma}^{1/2}\mathbf{v}^*/\|\boldsymbol{\Sigma}^{1/2}\mathbf{v}^*\|_2$.

From Proposition 2.1, running Algorithm 2 on the normalized distribution $G$ with error parameter $\eta \leq 1/(2000 C_\mathrm{A}^2)$ and failure probability $\tau$ consumes $\mathrm{poly}(d)\log(1/\tau)$ unlabeled samples and $O(d\log(d/\tau))$ labeled samples from $G$. Since the rejection sampling procedure for $G$ accepts points with probability at least $\Omega(\epsilon)$ Therefore, it consumes $\mathrm{poly}(d, 1/\epsilon)\log(1/\tau)$ samples from $D$ and uses $O(d\log(d/\tau))$ additional label queries. Conditioned on the event that it succeeds (which happens with probability at least $1 - \tau$), it either (i) reports that the $\mathbf{x}$-marginal of $G$ is not standard Gaussian (ii) returns a unit vector $\mathbf{w}$ such that

$$\|\mathbf{w} - (\mathbf{v}^*)'\|_2 \leq C_\mathrm{A}\sqrt{\Pr_{(\mathbf{x},y)\sim G}[\mathrm{sign}((\mathbf{v}^*)' \cdot \mathbf{x}) \neq y] + \eta}.$$
$$\tag{16}$$

In case (i), we can directly report that $\mathbf{x}$-marginal of $D$ is not $\mathcal{N}(\mathbf{0}, \mathbf{I})$ since the $\mathbf{x}$-marginal of $G$ ought to be $\mathcal{N}(\mathbf{0}, \mathbf{I})$.

In case (ii), we claim that at least one of the following hold (a) $\mathbf{v}$ before the localized update is already good enough, i.e. $\|\mathbf{v}^* - \mathbf{v}\|_2 \leq 40000C_{\mathrm{A}}{}^2\mathrm{opt}$; (b) it holds $\|(\mathbf{v}^*)' - \mathbf{w}\|_2 \leq 1/100$. Suppose that (a) does not hold; we will show that it then must hold $\|(\mathbf{v}^*)' - \mathbf{w}\|_2 \leq 1/100$. Since $\|\mathbf{v}^* - \mathbf{v}\|_2 \leq \delta$ and $\|\mathbf{v}^* - \mathbf{v}\|_2 > 40000C_{\mathrm{A}}{}^2\mathrm{opt}$, we have that $\mathrm{opt} \leq \delta/(40000C_{\mathrm{A}}^2)$. Furthermore, from Equation (15) we have that $\mathbf{Pr}_{(\mathbf{x},y)\sim D_{\mathbf{v},\delta}}[\mathrm{sign}(\mathbf{v}^* \cdot \mathbf{x}) \neq y] \leq 2\mathrm{opt}/\delta$, it then follows that $\mathbf{Pr}_{(\mathbf{x},y)\sim G}[\mathrm{sign}((\mathbf{v}^*)' \cdot \mathbf{x}) \neq y] \leq 2\mathrm{opt}/\delta \leq 1/(20000C_{\mathrm{A}}^2)$. Substituting this into Equation (16) then gives

$$\|\mathbf{w} - (\mathbf{v}^*)'\|_2 \leq 1/100 \ .$$

Using our assumption that $\|\mathbf{v} - \mathbf{v}^*\|_2 \leq \delta < 1/100$, we can apply Lemma 3.3, which gives that

$$\left\| \frac{\boldsymbol{\Sigma}^{-1/2}\mathbf{w}}{\|\boldsymbol{\Sigma}^{-1/2}\mathbf{w}\|_2} - \mathbf{v}^* \right\|_2 \leq \delta/2 \ .$$

Hence, we set $\mathbf{v}^{(t+1)} = \left\| \frac{\boldsymbol{\Sigma}^{-1/2}\mathbf{w}}{\|\boldsymbol{\Sigma}^{-1/2}\mathbf{w}\|_2} \right\|_2$ and this completes the proof. $\qquad\square$

---

**Input:** Sample access to $D_{\mathbf{x}}$ over unlabeled examples and query access to the labels of the samples drawn; a list of $k$ halfspaces $\{h^{(i)}\}_{i=1}^k$; parameters $\tau, \epsilon > 0$.
**Output:** finds a halfspace $\tilde{h}$ whose error is nearly optimal among the list.
  1. Let $C$ be a sufficiently large constant.
  2. For $i, j = 1 \cdots k$
     (a) Take $N_1 := C \ \log(k/\tau)/\epsilon$ samples $D_{\mathbf{x}}$.
     (b) Skip the iteration if there are fewer than $N_2 := 0.1 \ C \ \log(k/\tau)$ samples such that $h^{(i)}(\mathbf{x}) \neq h^{(j)}(\mathbf{x})$. Denote the samples satisfying the condition as $S_1$.
     (c) Query the labels of a random subset of samples from $S_1$ of size $N_2$. Denote the set of labeled samples as $S_2$
     (d) Compute the empirical errors of $h^{(i)}, h^{(j)}$ using $S_2$.
     (e) If the empirical errors differ by more than $1/4$, mark the one with larger error as sub-optimal.
  3. Return any halfspace among the list that has not been marked as sub-optimal or declare failure if there is not any.

**Algorithm 4:** Tournament

---

*Proof of Lemma 3.6.* Fix a halfspace $h^{(i)}$. Suppose there is another halfspace $h^{(j)}$ such that

$$\mathbf{Pr}_{(\mathbf{x},y)\sim D}[h^{(i)}(x) \neq y] > 10 \ \mathbf{Pr}_{(\mathbf{x},y)\sim D}[h^{(j)}(x) \neq y] + \epsilon.$$

We argue the halfspace will be marked as sub-optimal with probability at least $\tau/\log(1/\epsilon)$. For convenience, denote $a := \mathbf{Pr}_{(\mathbf{x},y)\sim D}[h^{(i)}(x) \neq y \wedge h^{(i)}(x) \neq h^{(j)}(x)]$, $b := \mathbf{Pr}_{(\mathbf{x},y)\sim D}[h^{(j)}(x) \neq y \wedge h^{(i)}(x) \neq h^{(j)}(x)]$ and $m := \mathbf{Pr}_{\mathbf{x}\sim D_{\mathbf{x}}}[h^{(i)}(x) \neq h^{(j)}]$. Notice that $a, b$ are the errors of the two halfspaces from the area in which their predictions differ and $m$ is the mass of the area. Then, it holds

$$a + b = \mathbf{Pr}_{(\mathbf{x},y)\sim D}[h^{(i)}(\mathbf{x}) \neq h^{(j)}(\mathbf{x}) \wedge y = h^{(i)}(\mathbf{x})] + \mathbf{Pr}_{(\mathbf{x},y)\sim D}[h^{(i)}(\mathbf{x}) \neq h^{(j)}(\mathbf{x}) \wedge y \neq h^{(i)}(\mathbf{x})] = m \ ,$$

$$a - b = \mathbf{Pr}_{(\mathbf{x},y)\sim D}\left[h^{(i)}(\mathbf{x}) \neq y\right] - \mathbf{Pr}_{(\mathbf{x},y)\sim D}\left[h^{(j)}(\mathbf{x}) \neq y\right] \geq 9 \ \mathbf{Pr}_{(\mathbf{x},y)\sim D}[h^{(j)}(x) \neq y] + \epsilon > 9b.$$

This further implies that $a/m - b/m > 4/5$. Notice that $a/m, b/m$ are precisely the errors of $h^{(i)}$, $h^{(j)}$ under the conditional distribution of $D$ restricted to the area in which $h^{(i)}(\mathbf{x}) \neq h^{(j)}(\mathbf{x})$. Hence,

with $\Theta(\log(k/\tau))$ many samples, the corresponding empirical errors will differ by at least $1/4$ with probability at least $\tau/k$.

On the other hand, suppose

$$\Pr_{(\mathbf{x},y)\sim D}[h^{(i)}(x) \neq y] \leq \Pr_{(\mathbf{x},y)\sim D}[h^{(j)}(x) \neq y].$$

This implies that $a/m < b/m$. As a result, the algorithm either skips the iteration if there are fewer than $\Theta(\log(k/\tau))$ many labeled samples from the conditional distribution or the empirical error of $h^{(i)}$ will not be less than that of $h^{(j)}$ by more than $1/4$ with probability at least $\tau/k^2$ if there are enough samples.

The correctness of the lemma then follows from union bound. Finally, we remark $m$, the mass of the area we condition to, should be at least $\epsilon$ if the errors of the two halfspaces differ by at least $\epsilon$. In that case, it takes on average $\Theta(1/\epsilon)$ unlabeled samples and 1 additional label query to simulate one labeled sample from the conditional distribution. Therefore, it suffices if the algorithm takes $O(k^2 \log(k/\tau)/\epsilon)$ many unlabeled samples from $D_{\mathbf{x}}$ and use $\Theta\left(k^2 \log(k/\tau)\right)$ many additional label queries. This concludes the proof of Lemma 3.6. $\qquad\square$

---

**Input:** Sample access to a distribution $D_{\mathbf{x}}$ over unlabeled examples and query access to the labels of the samples drawn; $\epsilon > 0$; unit vector $\mathbf{v} \in \mathbb{R}^d$.
**Output:** Either reports that $D_{\mathbf{x}}$ is not $\mathcal{N}(\mathbf{0}, \mathbf{I})$ or computes hypothesis $h$ such that $\Pr_{(\mathbf{x},y)\sim D}[h(\mathbf{x}) \neq y] = O(\text{opt})$.
1. Set $\tau = (\epsilon/(C\log(1/\epsilon)))$ for a sufficiently large constant $C > 0$. Set $\eta = 1/(20000C_{\mathrm{A}})$ where $C_{\mathrm{A}}$ is the constant from Proposition 2.1.
2. Run Algorithm 2 on $D$ with accuracy $\eta$ to obtain unit vector $\mathbf{v}^{(0)}$.
3. If Algorithm 2 reports that $D_{\mathbf{x}}$ is not $\mathcal{N}(\mathbf{0}, \mathbf{I})$, then report it and terminate.
4. For $t = 0 \ldots \log(1/\epsilon)$: set $\delta = (1/100)2^{-t}$ and run Algorithm 3 with parameters $\mathbf{v}^{(t)}, \delta, \eta$ to obtain $\mathbf{v}^{(t+1)}$.
5. For $i = 0 \ldots \log(1/\epsilon)$:
   (a) Run Algorithm 1 with $\mathbf{v}^{(i)}$ and $\eta = j\epsilon$ for $j \in [1/\epsilon]$ on the $\mathbf{x}$-marginals of $D$.
   (b) Set $h^{(i)}(\mathbf{x}) = \text{sign}(\mathbf{v}^{(i)} \cdot \mathbf{x})$.
6. Return the hypothesis obtained from running Algorithm 4 on the list of hypothesis $\{h^{(i)}\}_{i=1}^{\log(1/\epsilon)}$ on $D$ with accuracy $\epsilon$ and failure probability $\tau/10$.

**Algorithm 5:** Testable Localization

*Proof of Theorem 1.3.* Denote by $\mathbf{v}^*$, a unit vector with error at most opt, i.e., $\Pr_{(\mathbf{x},y)\sim D}[\text{sign}(\mathbf{v}^* \cdot \mathbf{x}) \neq y] \leq \text{opt}$. We start by analyzing Algorithm 5. In Line 2, Algorithm 5 uses Algorithm 2 with parameter $\eta = 1/(20000C_{\mathrm{A}}^2)$ to get a hypothesis $\mathbf{v}^{(0)}$ with small distance with $\mathbf{v}^*$. From Proposition 2.1, Algorithm 2 either reports that $\mathbf{x}$-marginal of $D$ is not $\mathcal{N}(\mathbf{0}, \mathbf{I})$ or outputs a vector $\mathbf{v}^{(0)}$ small distance with $\mathbf{v}^*$. If Algorithm 2 reports that the $\mathbf{x}$-marginal of $D$ is not $\mathcal{N}(\mathbf{0}, \mathbf{I})$, we can terminate the algorithm. Conditioned on the event that the algorithm did not terminate, then we have

$$\left\|\mathbf{v}^{(0)} - \mathbf{v}^*\right\|_2 \leq C_{\mathrm{A}}\sqrt{\text{opt} + \eta}. \tag{17}$$

We consider two cases depending on how large the value of opt is. If $\text{opt} > 1/(20000C_{\mathrm{A}}^2)$, then any unit vector achieves constant error; therefore, conditioned that the algorithm did not terminate on any proceeding test, any vector we output will satisfy the guarantees of Theorem 1.3. For the rest of the proof, we consider the case where $\text{opt} \leq 1/(20000C_{\mathrm{A}}^2)$. In this case, $\left\|\mathbf{v}^{(0)} - \mathbf{v}^*\right\|_2 \leq 1/100$, this means that Algorithm 5 on Lines 4-4 will decrease the distance between the current hypothesis and $\mathbf{v}^*$.

Conditioned on the event that the algorithm did not terminate at Lines 4-4 of Algorithm 5, we claim that there must have some $0 \leq t^* \leq \log(1/\epsilon)$ such that $\left\|\mathbf{v}^{(t^*)} - \mathbf{v}^*\right\|_2 \leq O\left(\text{opt} + \epsilon\right)$. Let $t' \in \mathbb{N}$ be

the maximum value so that $2^{-t'}/100 \geq 40000\text{opt}$, then, for all $t \leq \min(t', \log(1/\epsilon))$ it holds that $2^{-t}/100 \geq 40000\text{opt}$. From Lemma 3.5, we have that for all $t \leq \min(t', \log(1/\epsilon))$ it holds that

$$\left\| \mathbf{v}^* - \mathbf{v}^{(t)} \right\|_2 \leq 2^{-t-1}/100 .$$

From the above, note that if $t' > \log(1/\epsilon)$ then $\left\| \mathbf{v}^{(\log(1/\epsilon))} - \mathbf{v}^* \right\|_2 \leq \epsilon/100$. If $t' \leq \log(1/\epsilon)$, we have that $\left\| \mathbf{v}^{(t')} - \mathbf{v}^* \right\|_2 \leq O(\text{opt})$, which proves our claim. It remains to show that Algorithm 5 will return a vector $\mathbf{v}'$ so that $\mathbf{Pr}_{(\mathbf{x},y)\sim D}[\text{sign}(\mathbf{v}' \cdot \mathbf{x}) \neq y] = O(\text{opt} + \epsilon)$. From Lemma 3.1, conditioned that Algorithm 5 did not terminate on Lines 5-5b, we have that for all vectors $\mathbf{v}^{(0)}, \ldots, \mathbf{v}^{(\log(1/\epsilon))}$ generated on Lines 4-4 of Algorithm 5, we have that

$$\mathbf{Pr}_{(\mathbf{x},y)\sim D}[\text{sign}(\mathbf{v}^{(t)} \cdot \mathbf{x}) \neq y] \leq O\left( \left\| \mathbf{v}^{(t)} - \mathbf{v}^* \right\|_2 \right) .$$

Hence, we can conclude that $\mathbf{Pr}_{(\mathbf{x},y)\sim D}[\text{sign}(\mathbf{v}^{(t')} \cdot \mathbf{x}) \neq y] \leq O(\text{opt} + \epsilon)$. Then, applying Lemma 3.6 gives that the returned halfspace will have error at most $10 \cdot \mathbf{Pr}_{(\mathbf{x},y)\sim D}[\text{sign}(\mathbf{v}^{(t')} \cdot \mathbf{x}) \neq y] + \epsilon = O(\text{opt} + \epsilon)$.

To conclude the proof, note that each invocation of Algorithms 1, 2 and 3 consumes $\text{poly}(d, 1/\epsilon) \log(1/\tau)$ unlabeled samples and $O(d \ (\log(d/\tau) + \log\log(1/\epsilon)))$ label queries. Moreover, the total number of invocations is upper bounded by $\Theta(\log(1/\epsilon))$. For algorithm Algorithm 4, it draws $\text{poly}(1/\epsilon)$ many unlabeled samples and use $\Theta\left(\log^2(1/\epsilon) \ (\log\log(1/\epsilon) + \log(1/\tau))\right)$ label queries. Therefore the total unlabeled samples consumed by Algorithm 5 is $N = \text{poly}(d, 1/\epsilon)$ and the total label quereis used is at most

$$q = O\left( d \ \log(1/\epsilon) \ (\log(d/\tau) + \log\log(1/\epsilon)) + \log^2(1/\epsilon) \ (\log\log(1/\epsilon) + \log(1/\tau)) \right) .$$

All the procedures run in time that scales polynomially with the number of samples drawn. This concludes the proof. $\qquad\square$

