# OpenReview forum: "Efficient Testable Learning of Halfspaces with Adversarial Label Noise"
_NeurIPS.cc/2023/Conference — NeurIPS 2023 poster_

### Official Review · Reviewer_XEN1 · 2023-07-05

**Soundness:** 4 excellent
**Presentation:** 3 good
**Contribution:** 3 good
**Rating:** 8
**Confidence:** 3

**Summary:**

This paper studies the problem of learning halfspaces under Gaussian distribution and adversarial label noise in the testable learning framework of Rubinfeld and Vasilyan (STOC'23). In this setup, the learning algorithm, given a few training examples drawn i.i.d. from some unknown distribution $\mathcal{D}$, may either *accept* and returns a hypothesis or *reject*. The goal is to satisfy the following two natural constraints:
- Completeness: If the marginal of $\mathcal{D}$ is indeed Gaussian, the algorithm should accept w.h.p.
- Soundness: The probability that the learning algorithm accepts and returns an inaccurate hypothesis is small.

The main result is a poly-time algorithm that works for the Gaussian distribution against adversarial label noise. In particular, if $\mathsf{opt}$ is the smallest possible testing error achieved by the best halfspace on $\mathcal{D}$, the algorithm guarantees $O(\mathsf{opt}) + \epsilon$ error in time $\mathrm{poly}(d/\epsilon)$. Prior work achieves $\mathsf{opt} + \epsilon$ error but requires $d^{\mathrm{poly}(1/\epsilon)}$ time.

A first building block is a weak learner (given in Proposition 2.1) that outputs a weight vector $v$ that is $O(\sqrt{\mathsf{opt} + \eta})$-close in time $d^{\tilde O(1/\eta^2)}$. (This is not sufficient due to the extra square root as well as the exponential dependence on $1/\eta$.) This baseline learner is then boosted by localizing to the decision boundary of $\mathrm{sgn}(v\cdot x)$. In particular, each iteration either makes progress in improving the accuracy of $v$, or obtains a proof of non-Gaussianity in terms of moments.

**Strengths:**

The paper studies a well-motivated and fundamental learning theory problem in the recently proposed model of testable learning. The results are strong, and the solution involves several novel and non-trivial ideas. Despite this complexity, the main paper is nicely written and achieves a great balance between intuition and technical details. I vote for accepting this paper.

**Weaknesses:**

The algorithm only handles the homogeneous halfspaces without a bias term, and cannot achieve the $\mathsf{opt} + \epsilon$ error guarantee as in the prior work of [RV22, GKK22].

**Questions:**

Following up on the weakness part, some discussion on whether/how the current technique could handle non-homogeneous halfspaces and/or the stronger agnostic learning guarantee would be helpful.

**Limitations:**

This work is purely theoretical and its limitations are in the assumptions (e.g., the concept class is the class of homogeneous halfspaces, and the "reference distribution" is a spherical Gaussian). This has been properly addressed in the paper (clearly stated in the abstract and the main paper).

---

> ### Author Rebuttal · Authors · 2023-08-10
>
> We would like to thank the reviewer for the positive assessment of our work.
>
> First, regarding the limitation on the error guarantees (does not achieve OPT but only O(OPT)), we remark there is strong evidence that achieving error $OPT + \epsilon$ in fully polynomial time under agnostic learning, even without the testable requirement, is hard. In particular, as shown in [DKPZ21], no statistical query algorithm can learn halfspaces under Gaussian marginal up to error $OPT +\epsilon$ in time less than $d^{O(1/\epsilon^2)}$. Moreover, the same result is shown to hold for general algorithms under standard cryptographic assumptions in [DKR23]. Given the aforementioned hardness results, the best one can hope to do with a polynomial time algorithm is to learn up to error $O(OPT)$. Our result is optimal in a sense that it achieves essentially the same error guarantee efficiently as in the absence of the testable requirements.
>
> Secondly, the reviewer raises the question whether our approach can be made to work with non-homogeneous halfspaces. We remark that, to the best of our knowledge, almost all prior literature for learning LTFs under adversarial noise is for homogeneous cases with only two exceptions [DKS18, DKTZ22]. We believe our approach can be extended for general LTFs but would give a bit worse error. We leave this as an interesting question for future work.
>
> [DKS18] Diakonikolas, I., Kane, D. M., & Stewart, A. (2018, June). Learning geometric concepts with nasty noise. In Proceedings of the 50th Annual ACM SIGACT Symposium on Theory of Computing (pp. 1061-1073).
>
> [DKPZ21] Diakonikolas, I., Kane, D. M., Pittas, T., & Zarifis, N. (2021, July). The optimality of polynomial regression for agnostic learning under gaussian marginals in the SQ model. In Conference on Learning Theory (pp. 1552-1584). PMLR.
>
> [DKTZ22] Diakonikolas, I., Kontonis, V., Tzamos, C., & Zarifis, N. (2022, June). Learning general halfspaces with adversarial label noise via online gradient descent. In International Conference on Machine Learning (pp. 5118-5141). PMLR.
>
> [DKR23] Diakonikolas, I., Kane, D., & Ren, L. (2023, July). Near-optimal cryptographic hardness of agnostically learning halfspaces and relu regression under gaussian marginals. In International Conference on Machine Learning (pp. 7922-7938). PMLR.

---

> > ### Comment · Reviewer_XEN1 · 2023-08-14
> >
> > Thank you for answering my questions! My overall evaluation of the paper remains positive.

---

### Official Review · Reviewer_1kSW · 2023-07-06

**Soundness:** 2 fair
**Presentation:** 3 good
**Contribution:** 3 good
**Rating:** 4
**Confidence:** 4

**Summary:**

Learning halfspaces is a very well studied problem in machine learning. In the agnostic (adversarial label noise) and distribution free setting, this problem has been known to be computationally intractable. As a result, there have been several works of agnostic learning in distribution specific settings (where the marginal distribution belongs to a particular family of distributions, say Gaussian or log-concave). In this scenario, the learner has an error of the form $OPT + \epsilon$, where OPT denotes the optimal 0-1 error. This however has complexity $d^{1/\epsilon^2}$, and the exponential dependency on $1/\eps$ is tight. This motivates the designing of learning algorithms that have better sample complexity with respect to $1/\epsilon$, whereas the error becomes $f(OPT) + \eps$ for some function f.

In this work, the authors studied this problem in the newly introduced Testable learning framework  by Rubinfeld and Vasilyan (STOC 23). Here the goal is if the tester accepts, then with high probability the output of the learning is close to some function of OPT, and if the data satisfies the distributional assumptions, the algorithm accepts.

In this paper, the authors study the class of homogeneous halfspaces over $R^d$ with Gaussian marginals. This work designs a tester-learner with sample complexity $N=poly(d, 1/\epsilon)$ and runtime $poly(dN)$. Additionally, their algorithm also works in the active learning framework where the algorithm only queries the labels of a subset of the samples.

The authors first design a weak agnostic learner with respect to Gaussian distribution which checks if the low degree moments of the marginal distribution approximately matches with that of Gaussian distribution (Section 2). The idea is if the moments are close, then the vector corresponding to the degree-1 chow parameters will not be far from v^* (the true vector of the half-space), see Lemma 2.3. They also design an algorithm that either reports if the marginal distribution is not Gaussian or it outputs a unit vector w (Algorithm 2 corresponding to Proposition 2.1)  Once they have the weak learner, the authors will try to find a vector that has small 0-1 error with v^* by calling Algorithm 2 (Lemma 2.1) in an iterative manner. After logarithmic many iterations, they show that the probability mass of the disagreement region is bounded (Lemma 3.1). However, this process gives a collection of vectors such that one of them is close to the optimal vector v^*.The authors finally run a tournament among these candidate halfspaces corresponding to these vectors and outputs the winner hypothesis (Lemma 3.6).




**Strengths:**

The paper gives the first algorithm for testable learning of halfspaces that runs in poly(d, epsilon). The algorithm is very nice.
With the complexity pulled down drastically, a proper implementation and experimental results for this algorithm would be possible and it would be nice to see the relevance of the concept of testable learning in various applications.

**Weaknesses:**

The usefulness of the testable learning model in real life applications is yet to be understood.  The paper can only handle adversarial noise.


**Questions:**

Can this approach be used to design tester-learners for function classes other than halfspaces.


**Limitations:**

It is a pure theoretical work in the paradigm of testable learning - a relatively new concept whose importance is not yet fully confirmed.

---

> ### Author Rebuttal · Authors · 2023-08-10
>
>
> Regarding practicality: The testable learning framework is a new learning paradigm whose definition (in a paper that appeared in STOC’23) was motivated by practical considerations (namely, the fact that one cannot verify the output of known distribution-specific agnostic learners). Since its introduction, the ML theory community has been intensely studying the algorithmic possibilities and limitations in this learning model. Ours is the first work that gives fully-polynomial time algorithms with near-optimal error guarantees for learning Gaussian halfspaces in this model. Since the testable learning model was defined only recently, it has not yet been picked up by more practical ML researchers. Related to this, we note that prior to our work no known efficient implementation of testable learners was possible (because prior algorithms had complexity scaling exponential in $1/\epsilon$). As pointed out by the reviewer, our work drastically pulls down the complexity and hence makes experimental investigation of the model in practical settings a viable option.
>
> Finally, we would like to note that learning theory results such as ours are within the scope of NeurIPS (as stated in the call for papers). Consequently, we request that our work is judged on its merits based on the appropriate criteria.
>
> Regarding generalizing our results to other hypothesis classes: even without the testable requirement, halfspaces are the only class for which it is known how to achieve O(OPT)+eps error in polynomial time in the distribution-specific agnostic model. In fact, most known efficient learners (with distribution-independent error guarantees) tolerant to label noise focus on the class of halfspaces and their generalizations [KKMS08, KLS09, ABL17, DKS18, DKKTZ21, DKPTZ21, DKTZ22].
>
>
> [ABL17] Awasthi, P., Balcan, M. F., & Long, P. M. (2017). The power of localization for efficiently learning linear separators with noise. Journal of the ACM (JACM), 63(6), 1-27.
>
> [DKS18] Diakonikolas, I., Kane, D. M., & Stewart, A. (2018, June). Learning geometric concepts with nasty noise. In Proceedings of the 50th Annual ACM SIGACT Symposium on Theory of Computing (pp. 1061-1073).
>
> [DKKTZ21] Diakonikolas, I., Kane, D. M., Kontonis, V., Tzamos, C., & Zarifis, N. (2021, July). Agnostic proper learning of halfspaces under gaussian marginals. In Conference on Learning Theory (pp. 1522-1551). PMLR.
>
> [DKPZ21] Diakonikolas, I., Kane, D. M., Pittas, T., & Zarifis, N. (2021, July). The optimality of polynomial regression for agnostic learning under gaussian marginals in the SQ model. In Conference on Learning Theory (pp. 1552-1584). PMLR.
>
> [KKMS08] Kalai, A. T., Klivans, A. R., Mansour, Y., & Servedio, R. A. (2008). Agnostically learning halfspaces. SIAM Journal on Computing, 37(6), 1777-1805.
>
> [KLS09] Klivans, A. R., Long, P. M., & Servedio, R. A. (2009). Learning Halfspaces with Malicious Noise. Journal of Machine Learning Research, 10(12).
>
> [DKTZ22] Diakonikolas, I., Kontonis, V., Tzamos, C., & Zarifis, N. (2022, June). Learning general halfspaces with adversarial label noise via online gradient descent. In International Conference on Machine Learning (pp. 5118-5141). PMLR.

---

### Official Review · Reviewer_8SDJ · 2023-07-09

**Soundness:** 4 excellent
**Presentation:** 4 excellent
**Contribution:** 3 good
**Rating:** 7
**Confidence:** 3

**Summary:**

This paper considers computationally efficient learning of halfspaces with adversarial noise in a recently proposed "testable" task: with high probability either reports the example distribution is not standard Guaissian, or outputs a halfspace with error of at most O(opt)+epsilon. There are existing works in a similar setting but targets the error of opt+epsilon, so their time and sample complexity is unavoidably $d^{O(1/\epsilon^2)}$. This paper relaxed the error target to O(opt)+epsilon as in previous non-testable works, and shows that the testable task can be achieved with time and sample complexity of around poly(d, 1/epsilon) and label complexity of around d log(1/epsilon).

Technically, the results are obtained by (non-trivially) combining known ideas, like moment matching to assist testing, the Chow parameter (roughly speaking, E[yx]), robust mean estimation, and soft localization for learning in a label-efficient way.

--

I've read the author's response and I remain positive for this paper.

**Strengths:**

- This paper considers a recently proposed task of efficient testable learning of halfspaces with Gaussian distribution, and shows that, similar to the results in classical non-testable setting, it can be done with poly(d/epsilon) when considering constant-factor approximation. In my opinion, this result is not very surprising, but it is still interesting to know.

- Most of the individual ideas used for the proof and algorithm are known to the community. On the other hand, the careful analysis to combine them in this new testable learning setting is still novel and nontrivial.

- I briefly checked the proofs and they looked sound to me.

- The paper is written very clearly: it clearly explains the problem and how it relates to existing work, and it organizes its technical part well so that one could follow both the high level steps and details.

**Weaknesses:**

There are some weakensses (and I mentioned some of them above already), but I don't think they're really major:

- The results are not very surprising to me, and the individual ideas used are mostly known.

- It only considers standard Gaussian distributions and adversarial noise.

**Questions:**

I don't really have any questions, but one minor comment is that since this is a relatively new setting, it might help readers understand the difficulty of the problem by explaining that one cannot simply first test if the distribution is Gaussian and then proceed with standard learning algorithms.

---

> ### Author Rebuttal · Authors · 2023-08-09
>
> The reviewer mentions the result is itself not surprising. While we respect this opinion, we want to point out that there are known hypotheses classes where there are separations between testable learning and standard agnostic learning. In particular, as shown in the initial work of [RV23], there are agnostic learning algorithms of monotone boolean functions under the uniform distribution in time $2^{\tilde O(\sqrt n)}$, but their lower bound construction suggest that achieving the same under the testable learning setting requires a runtime of $2^{\Omega(n)}$.
>
> The reviewer also pointed out “Most of the individual ideas used for the proof and algorithm are known to the community”. While we indeed use many standard tools appearing in prior works, the wedge bound algorithm (Algorithm 2) and its analysis, to the best of our knowledge, is a novel technical contribution. Quite differently from the tester that exists in prior literature, this testing routine no longer blindly compares global low-degree moments but rather uses “local’’ information restricted to areas defined by the output of the learner. In our opinion, this is one of the most important pieces in achieving efficient testable learning of halfspaces.
>
> Finally, we thank the reviewer for bringing up the discussion of the difficulty of testable learning. The insufficiency of the naive combination of testing distributional closeness and the standard learning algorithm has already been discussed in the initial paper of [RV23]. We will include this in our paper and explain in detail why testable learning is a non-trivial task.
>
> [RV23] Rubinfeld, R., & Vasilyan, A. (2023, June). Testing distributional assumptions of learning algorithms. In Proceedings of the 55th Annual ACM Symposium on Theory of Computing (pp. 1643-1656).

---

### Official Review · Reviewer_fkEL · 2023-07-09

**Soundness:** 3 good
**Presentation:** 3 good
**Contribution:** 3 good
**Rating:** 7
**Confidence:** 4

**Summary:**

The paper studies the problem of efficient testable learning of half spaces under Gaussian marginals and adversarial noise. The setting of testable learning addresses the issue of restrictive assumptions on the marginal distributions (such as Gaussianity) under learning algorithms are designed. Traditionally, a learning algorithm has no guarantee when the marginal distribution is not the prescribed one. In tester-learner setting, a learning algorithm consists of a tester and a learner. The tester "certifies" that the marginal distribution satisfies certain conditions (ideally this test would be a test of whether the marginal distribution is "close" to the original distribution). Conditioned on the tester passing, the learning algorithm has the desired correctness guarantee. The main idea is that tester does not need to test for distributional closeness but rather just closeness sufficient to "fool" the learning algorithm.

A recent line of work has considered the learning of half spaces in this setting. Previous work show that the halfspaces can be agnostically learnt under Gaussian marginals in time $d^{1/\eps^2}$ which is optimal if one insists on algorithms that return hypotheses that have error $OPT + \eps$. The present paper relaxes this notion to require that the hypothesis returned only has error $O(OPT) + \eps$. The paper presents a polynomial (in $d$ and $1/\eps$) time tester-learner algorithm under Gaussian marginals.

**Strengths:**

The paper presents a natural and interesting adaptation of the localization technique for learning half spaces in the testable learning setting. The main idea is that learning algorithms designed to work under Gaussian marginals require significantly less information (for example only rely on moments and anti concentration). The paper analyses a robust version of Chow parameter recovery under the condition that the marginals match Gaussian marginals (this can be tested efficiently). Using this as a weak learner, the paper uses a localization technique to convert a weak learner (from Chow parameter recovery) to a strong learner. Again the paper uses the fact that closeness of parameter and loss for halfspaces can be efficiently tested. In summary, the paper presents a nice analysis of learning algorithm

**Weaknesses:**

As presented the paper is too specific to Gaussian marginals.

**Questions:**

- It would be nice to address which parts of the paper can be generalized to more general distributions? For example, what notions of testable concentration and anti concentrations would suffice for these algorithms to work. For example, would having sufficient moment decay suffice for the Chow parameter recovery?

**Limitations:**

Yes

---

> ### Author Rebuttal · Authors · 2023-08-09
>
>
> Regarding the concern that our tester-learner may only work under Gaussian marginals, we refer the reviewer to our General Response.
>
> The reviewer brought up the question of which part of the algorithm can be generalized to work under a broader family of distributions. Just as the reviewer guessed, the Chow parameter estimation part can indeed be relaxed to work with distributions whose second moments are bounded. The main challenge in generalizing our approach to other distributions is the localization step: while the conditional distribution of Gaussian restricted to a thin slice is still a Gaussian, this need not be the case for other family of distributions.

---

> > ### Comment · Reviewer_fkEL · 2023-08-18
> >
> > We thank the authors for the response and maintain my positive score.

---

### Author Rebuttal · Authors · 2023-08-09

## General Response

We thank the reviewers for their time and effort in providing feedback. We are encouraged by the positive comments from reviewers for the following: (i) the **improved running time for testable learning halfspaces** from $d^{ \text{poly}(1/\epsilon) }$ to fully polynomial in all parameters (ii) the **optimality of the error guarantees** (Reviewers fkEL, 8SDJ, XEN1) and (iii) **the writing quality and the clarity of the presentation of the ideas** (Reviewers 8SDJ, XEN1). Additionally, as mentioned by reviewer 1kSW, we emphasize that another major strength of our algorithm is that it **works under the active learning setting requiring only $d \text{ } \text{poly} \log(1/\epsilon)$ labeled examples.**

Below, we first address some common concerns and then provide responses to each individual reviewer in order.

**Learning under Gaussian Marginals**

 Two reviewers pointed out that our result is specific for Gaussian marginals. While generalizing our results to broader families of distributions is an interesting future direction, we remark that Gaussianity here is a natural assumption that has been commonly made in many prior works studying agnostic learning of halfspaces (see [DKKTZ21], [DKPTZ21], [DKZ20], etc.). Analogous to situations in the standard agnostic learning setting, understanding polynomial time testable learners in the Gaussian case can be seen as an important first step in achieving a universal theory of efficient testable learning under a variety of different distributions. We emphasize that our work is the first polynomial-time algorithm for testable learning halfspaces with near-optimal error guarantees that matches that without the testable requirement. Therefore, we view this is a natural and important first step in understanding efficient testable learning under more general marginal distributions.

Some reviewers also questioned the choice of our noise model. We remark that the adversarial noise model is theoretically well motivated, extensively studied, and also one of the strongest noise models in the literature. Hence, the error guarantees and runtime hold for all other weaker families of noise models as well. Whether the error guarantees can be improved under these weaker noise conditions is a direction we did not pursue but is a potential interesting future direction.

**Adversarial Noise Model**

Some reviewers also questioned the choice of our noise model. We remark that the adversarial noise model is theoretically well motivated, extensively studied, and also one of the strongest noise models in the literature. Hence, the error guarantees and runtime hold for all other weaker families of noise models as well. Whether the error guarantees can be improved under these weaker noise conditions is a direction we did not pursue but is a potential interesting future direction.

---

### Decision · Program_Chairs · 2023-09-21

**Decision:**

Accept (poster)

**Comment:**

This paper studies the recently introduced testable learning model and studies the problem of robustly learning a halfspace. The paper presents a polynomial time algorithm that achieves the near optimal $O(OPT) + \epsilon$ guarantee. The reviewers liked the paper and concluded that the result is a solid theoretical contribution. One reviewer questioned the applicability of the testable learning model in real world settings and wondered whether the paper would be more suitable for a TCS venue. In my assessment, the paper fits well into the scope of NeurIPS. The authors are however encouraged to improve the writing and discuss more practical applications (both current and forward looking) of the model.